# Evolving LLMs' Self-Refinement Capability via Synergistic Training-Inference Optimization

## Abstract

*Self-Refinement* refers to a model's ability to revise its own responses to produce improved outputs. This capability can also serve as a fundamental mechanism for *Self-Improvement*, for example by reconstructing datasets with refined results to enhance intrinsic model performance. However, our comprehensive experiments reveal that large language models (LLMs) show no clear evidence of inherent *Self-Refinement*; on average, response quality degrades over successive iterations. To address this gap, we propose **EVOLVE**, a simple yet effective framework for eliciting and tracking the evolution of *Self-Refinement* through iterative training. Moreover, we demonstrate the potential of leveraging *Self-Refinement* to achieve broader *Self-Improvement* of intrinsic model abilities. Experiments show that the evolved *Self-Refinement* ability enables the Llama-3.1-8B base model to surpass GPT-4o, achieving 62.3% length-controlled and 63.3% raw win rates on AlpacaEval 2, and 50.3% on Arena-Hard. It also generalizes effectively to out-of-domain reasoning tasks, improving performance on mathematical reasoning benchmarks such as GSM8K and MATH.

## 1 Introduction

Large language models (LLMs) have demonstrated strong performance across a wide range of tasks through training on massive datasets (Achiam et al., 2023; Dubey et al., 2024). However, the supply of high-quality training data is becoming increasingly scarce, limiting further progress. As a complementary direction, *Self-Improvement* methods seek to enhance a model's intrinsic capabilities by leveraging model-generated data and feedback with minimal external supervision (Tao et al., 2024; Huang et al., 2022). A critical challenge in this field is ensuring the quality and robustness of synthetic datasets while minimizing noise, often by leveraging intrinsic model mechanisms or properties. To this end, we investigate inference-time *Self-Refinement*, where a model revises its own outputs to improve response accuracy and stability. Previous work has investigated incorporating *Self-Refinement* strategies into broader frameworks, ranging from prompt-based techniques that let models iteratively revise their own drafts (Madaan et al., 2024; Paul et al., 2023) to approaches that incorporate external feedback, such as preference signals or additional contextual cues (Stiennon et al., 2020; Asai et al., 2024). These efforts raise a fundamental question: ***Can LLMs autonomously refine their responses without external information, thereby achieving Self-Refinement?***

To investigate this question, we conducted preliminary experiments to assess whether contemporary LLMs possess an inherent *Self-Refinement* capability. To mitigate potential biases from prompt design, we evaluated three distinct refinement templates (detailed in Appendix J), spanning a spectrum from detailed guidance to minimal intervention: (1) a guided template requiring direct output of an improved response, emphasizing clarity, accuracy, and conciseness without analysis; (2) a guided template mandating an initial analysis of the example response's strengths and weaknesses, followed by the refined output; and (3) a minimalist template with no explicit instructions, to minimize prompt-induced artifacts. These templates were applied across diverse LLMs with varying architectures, enabling iterative *Self-Refinement* cycles. As shown in Fig. 1, our results reveal no clear evidence of inherent *Self-Refinement*; on average, response quality degrades over successive iterations.

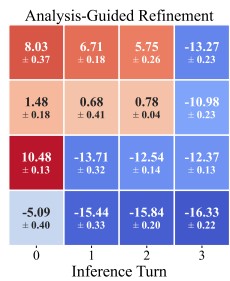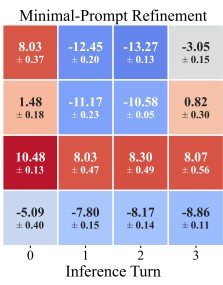

Figure 1: Evaluation of *Self-Refinement* Capability Across Various Models. We use three refinement templates to minimize prompt bias. The x-axis denotes the inference iteration number. For each turn, responses are generated from 256 UltraFeedback test set samples, using the original prompt and the prior turn's output. These are then scored by the Skywork Reward Model (Liu et al., 2024). To eliminate potential randomness, the reported values are the mean score of three independent runs with different random seeds; higher scores indicate better quality. Templates are detailed in Appendix J.

Motivated by these findings, we move beyond diagnosis to ask: *can this capability be effectively activated and strengthened through training, and how might it be leveraged for sustained Self-Improvement?* Specifically, we first investigate fine-tuning methods to activate the *Self-Refinement* capabilities of LLMs. Building on this, we further explore strategies to continuously enhance LLMs' *Self-Refinement* abilities, encompassing ongoing data acquisition and iterative training updates. To analyze the evolution of *Self-Refinement* capabilities in stages, we conduct our study within an iterative preference training framework. Finally, we examine potential applications of activated *Self-Refinement* capability, such as leveraging it to achieve *Self-Improvement* of the model's intrinsic capabilities. The contributions of our work are summarized as follows:

- We conduct a comprehensive study on eliciting and enhancing *Self-Refinement* capability through iterative training, including proposing a novel training method to generate improved responses from previous outputs, and exploring diverse strategies for effectively leveraging *Self-Refinement* to collect datasets.

- We introduce EVOLVE, a simple yet effective framework for analyzing the evolution of *Self-Refinement* ability throughout iterative training cycles. Furthermore, we investigate the potential of leveraging *Self-Refinement* to achieve *Self-Improvement* of model intrinsic abilities.

- We empirically validate the effectiveness of our framework in enhancing *Self-Refinement* capability. Starting from the Llama-3.1-8B base model, EVOLVE outperforms Llama-3.1-405B-Instruct and GPT-4o, achieving 62.3% length-controlled and 63.3% raw win rates on AlpacaEval 2, along with 50.3% on Arena-Hard, while also improving performance on mathematical reasoning benchmarks such as GSM8K and MATH.

## 2 RELATED WORKS

**LLM Self-Improvement.** *Self-Improvement* in LLMs aims to enhance intrinsic model capabilities with minimal external supervision, addressing the challenge of scarce high-quality training data (Tao et al., 2024; Huang et al., 2022). Recent approaches leverage model-generated data to iteratively improve model performance, often through synthetic dataset construction or feedback-driven optimization (Wang et al., 2022a;b; 2025). For instance, self-consistency decoding (Wang et al., 2022a) enables models to sample multiple outputs and select the most consistent, improving performance in reasoning tasks, while methods like Self-Instruct (Wang et al., 2022b) automate instruction-tuning data creation to enhance task generalization. Other works explore iterative bootstrapping, where models refine their outputs over multiple rounds (Madaan et al., 2024; Huang et al., 2022), or employ internal scoring mechanisms to filter and improve dataset quality (Yuan et al., 2024). Our work builds on these foundations, focusing on activating and enhancing *Self-Refinement* as a mechanism for sustained *Self-Improvement*, distinct from prior approaches by emphasizing iterative training to strengthen intrinsic refinement capabilities.

**Data Generation and Iterative Optimization.** The scarcity of high-quality training data has motivated the development of new data-generation strategies to enable sustainable iterative optimization (Long et al., 2024; Ding et al., 2024). Techniques such as data augmentation through LLM-generated synthetic datasets have proven effective in scaling training data for smaller models (Chen et al., 2023; Xu et al., 2024; Taori et al., 2023). Recent studies further explore online data generation, where

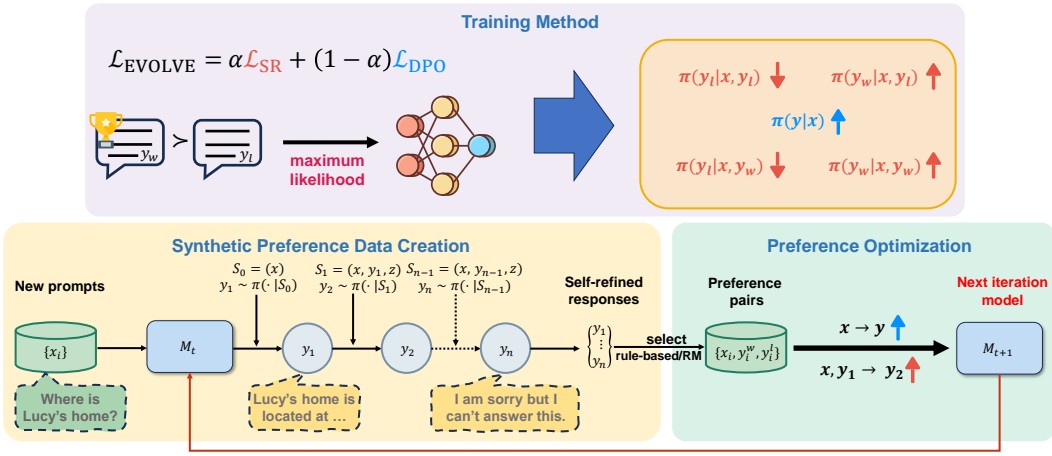

Figure 2: Our framework, **EVOLVE**, iteratively alternates between inference and training processes. In iteration $t$, Model $M_t$ uses the *Self-Refinement* strategy to collect preference data, which is then utilized to enhance the model's intrinsic capabilities via preference training (Eq. 5), yielding the next iteration model $M_{t+1}$. The dataset is filtered through either a rule-based method or a reward model.

models dynamically generate and refine datasets during training (Tian et al., 2024; Xiong et al., 2023). In this work, we develop a closed-loop system that harnesses models' *Self-Refinement* capabilities to autonomously generate graded data, enabling iterative model optimization.

## 3 METHODOLOGY

In this section, we first investigate the training method for activating *Self-Refinement* capabilities in LLMs. Building on the activated *Self-Refinement* abilities, we further investigate approaches to progressively enhance the *Self-Refinement* capacities of LLMs, encompassing both data acquisition and iterative training components. Drawing on these findings, we propose a simple yet effective framework EVOLVE for iterative training and inference to study the evolution of *Self-Refinement* capabilities during the iterative training process.

**Evaluation Protocol.** To substantiate our analysis below, we now describe the evaluation protocol employed in this section. For each generation round, we use 256 samples from the UltraFeedback (Lambert et al., 2024a) test set to generate responses. These responses are then scored using the Skywork Reward Model (Liu et al., 2024), a compact yet high-performing model on the RewardBench leaderboard (Lambert et al., 2024b). Results are reported as average scores.

### 3.1 TRAINING STAGE OF EVOLVE: SYNERGISTIC OPTIMIZATION OF SFT AND PT

Starting from a base pretrained language model, we investigate post-training fine-tuning methods, covering a synergistic optimization process of Supervised Fine-Tuning (SFT) and Preference Training (PT) stages, to activate and enhance the *Self-Refinement* capability of LLMs. Specifically, SFT serves to initiate *Self-Refinement*, enabling the model to learn how to revise suboptimal responses, while PT further strengthens this ability, improving the model's proficiency in applying it during iterative optimization. These two stages complement each other, forming the foundation of our framework. To support this claim, we first present the experimental results of the different training configurations tested for the SFT and PT stages, as shown in Fig. 3.

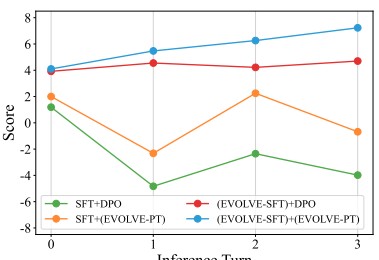

Figure 3: Ablation of training combinations. SFT activates *Self-Refinement*, PT enhances it, and their synergy (blue, ours) yields the best performance.

**Supervised Fine-Tuning (SFT) for Activating Self-Refinement.** Our experimental findings reveal that the SFT stage is fundamental for instilling the *Self-Refinement* capability, enabling the effective operation of the entire framework. Building on the standard Negative Log-Likelihood (NLL) SFT loss, we introduce specific modifications to explicitly encourage the model to refine suboptimal

responses:

$$\mathcal{L}_{\text{EVOLVE-SFT}}(\pi_\theta) = -\underset{(x,y_w,y_l)\sim\mathcal{D}}{\mathbb{E}}[\log \pi_\theta(y_w|x) + \log \pi_\theta(y_w|x,y_l,z)], \quad (1)$$

where $(x, y_w, y_l)$ is sampled from a preference dataset $\mathcal{D}$, $x$ represents the prompt, $y_w$ is the preferred response, and $y_l$ is the rejected response. The variable $z$ denotes the refinement template, as shown in Appendix D.1.

Without these SFT adaptations, the model fails to adequately learn *Self-Refinement* behavior. This is reflected in Fig. 3, where the response quality oscillates significantly across inference turns.

**Preference Training (PT) for Strengthening Self-Refinement.** PT stage is designed to further enhance the model's *Self-Refinement* capability based on the SFT phase, enabling it to more effectively master and apply this skill. To improve the model's ability to refine an initial response into a better one during PT, we first consider the following scenario: given a prompt $x$ and an initial response $y_1$, how can the model be guided to produce a refined response $y_2$? We model this problem as follows:

$$\max_\pi \underset{y_2\sim\pi(\cdot|x,y_1,z)}{\mathbb{E}}\left[p(y_2 \succ y_1|x) - \beta D_{\text{KL}}(\pi||\pi_{\text{ref}}|x,y_1,z)\right]. \quad (2)$$

where $p(y_2 \succ y_1|x)$ is the preference function, denoting the probability that $y_2$ is preferred over $y_1$ given $x$. By deriving the optimal policy from Eq. 2 and applying a mean squared error formulation, we parameterize the policy model as $\pi_\theta$, yielding:

$$\mathcal{L}(\pi_\theta;\pi_{\text{ref}}) = \underset{(x,y_1,y_2)\sim\rho}{\mathbb{E}}\left[\log\left(\frac{\pi_\theta(y_2|x,y_1,z)\pi_{\text{ref}}(y_1|x,y_1,z)}{\pi_\theta(y_1|x,y_1,z)\pi_{\text{ref}}(y_2|x,y_1,z)}\right) - \left(\frac{p(y_2\succ y_1|x)}{\beta} - \frac{1}{2\beta}\right)\right]^2, \quad (3)$$

where $\rho$ denotes the true distribution (derivation in Appendix C.1). Given a curated preference dataset $\mathcal{D} = \{(x^{(i)}, y_w^{(i)}, y_l^{(i)})\}_{i=1}^N$ with $y_w \succ y_l$, we further define the final *Self-Refinement* loss:

$$\mathcal{L}_{\text{SR}}(\pi_\theta;\pi_{\text{ref}}) = \underset{(x,y_w,y_l)\sim\mathcal{D}}{\mathbb{E}}\left[\frac{1}{2} - v(x,y_l,y_w,z;\pi_\theta)\right]^2 + \underset{(x,y_w,y_l)\sim\mathcal{D}}{\mathbb{E}}\left[\frac{1}{2} + v(x,y_w,y_l,z;\pi_\theta)\right]^2, \quad (4)$$

where $v(x,y_1,y_2,z;\pi_\theta) = \beta\log\left(\frac{\pi_\theta(y_2|x,y_1,z)\pi_{\text{ref}}(y_1|x,y_1,z)}{\pi_\theta(y_1|x,y_1,z)\pi_{\text{ref}}(y_2|x,y_1,z)}\right)$. Finally, we combine the *Self-Refinement* loss with the DPO loss (Rafailov et al., 2023), obtaining:

$$\mathcal{L}_{\text{EVOLVE-PT}}(\pi_\theta;\pi_{\text{ref}}) = \alpha\mathcal{L}_{\text{SR}}(\pi_\theta;\pi_{\text{ref}}) + (1-\alpha)\mathcal{L}_{\text{DPO}}(\pi_\theta;\pi_{\text{ref}}). \quad (5)$$

The effectiveness of the PT stage is inherently tied to the foundational adjustments made during SFT, while the SFT phase requires the continuous activation of the *Self-Refinement* capability from the PT stage, demonstrating that **the two phases are truly complementary**. As shown in Fig. 3, without the refinements introduced in the PT stage, the *Self-Refinement* capability acquired during SFT remains limited. Thus, only through the synergistic optimization of both stages can the model's *Self-Refinement* ability be effectively activated and consistently strengthened. Furthermore, as discussed in Appendix E, we explore alternative *Self-Refinement* objective formulations in the PT stage, observing comparable improvements. This suggests that the development of *Self-Refinement* stems from inherent advantages of the overall training paradigm, rather than a particular loss design.

### 3.2 INFERENCE STAGE OF EVOLVE: EXPLORING DYNAMIC GENERATION STRATEGIES

Building on the training stage, we next examine how to exploit the acquired *Self-Refinement* capability during inference. To identify the most suitable strategy for our final framework, we systematically summarize and compare four distinct generation strategies:

- **Parallel Sampling**: Given a problem $x$, the model generates multiple candidate answers $\{y_1, y_2, \ldots, y_n\}$ independently in parallel.
- **Chain of Self-Refinement**: For a problem $x$, the model first generates an initial answer $y_1$. Using the refinement template $z$, it then iteratively produces refined responses $y_n$ based on $x$ and the previous response $y_{n-1}$.
- **Few-Shot Self-Refinement**: The model iteratively improves its outputs $y_n$ by conditioning on both the input $x$ and all prior generations $\{y_1, y_2, \ldots, y_{n-1}\}$.
- **Self-Refinement with Self-Evaluation**: The model first generates an initial response $y_1$ for the problem $x$, then evaluates it along multiple dimensions (e.g., relevance, helpfulness) to produce an evaluation $e_1$. At each subsequent step, a refined response $y_n$ is generated based on $x$, the previous response $y_{n-1}$, and the last evaluation $e_{n-1}$: $y_n \sim \pi_\theta(\cdot \mid x, y_{n-1}, e_{n-1})$. Each evaluation $e_n$ is generated from $(x, y_n)$.

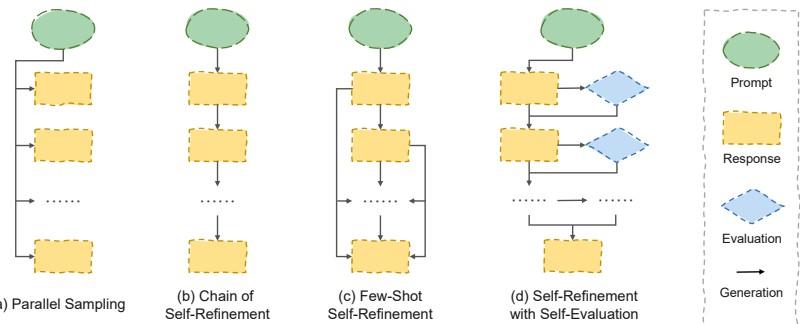

Figure 4: Illustration of four dynamic generation strategies.

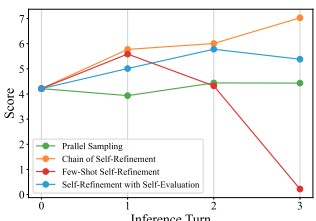

The procedural logic of four generation strategies is presented in Fig. 4 and corresponding prompt templates are provided in Appendix K. Based on the experimental results shown in Fig. 5, we can make the following observations: *Parallel Sampling* maintains stable mean rewards across iterations, as all outputs are drawn from the same distribution. *Chain of Self-Refinement* achieves the best performance, consistently improving response quality across iterations. *Few-Shot Self-Refinement* exhibits an initial increase but eventually degrades, as longer prompts hinder the model's ability to capture key information. *Self-Refinement with Self-Evaluation* also yields iterative gains but remains weaker than Chain of Self-Refinement, primarily due to the model's limited self-evaluation capability, as illustrated in Appendix K. Since **Chain of Self-Refinement best aligns with our training approach, consistently improves response quality, and incurs only minimal overhead**, we adopt it as the core generation strategy in our framework.

Figure 5: Performance of four generation strategies. *Chain of Self-Refinement* achieves the best results across iterations.

In summary, by integrating insights from both the training and inference phases, we propose EVOLVE, a streamlined and effective framework, as illustrated in Fig. 2. Our EVOLVE framework iteratively conducts preference training and data generation. This iterative training process continually enhances the LLMs' *Self-Refinement* capability, which in turn enables the hierarchical generation of high-quality data. Through this mutually reinforcing mechanism, EVOLVE achieves sustained improvement in a unified training loop.

## 4 EXPERIMENTS

Building on the evaluation in Fig. 1, where current LLMs show no clear evidence of inherent *Self-Refinement*, we propose the EVOLVE framework to enhance this capability. In this section, we demonstrate its effectiveness by addressing three key questions:

1. How much does EVOLVE improve model performance compared to prior methods (§4.2)?
2. Which components are responsible for the performance improvements of EVOLVE (§4.3)?
3. Does the *Self-Refinement* ability induced by EVOLVE generalize to out-of-domain tasks (§4.4)?

### 4.1 EXPERIMENTS SETTING

**Models and Training Settings.** Our experiments were conducted on the Llama-3.1-8B Base model (Dubey et al., 2024) and Mistral-7B Base model (Jiang et al., 2023). During the **SFT phase**, we fine-tune the base model on the llama-3.1-tulu-3-70b-preference-mixture dataset (Lambert et al., 2024a) using Eq. 1, yielding the EVOLVE-SFT model. In the **preference training phase**, we continue to train the EVOLVE-SFT model using 30K preference pairs from the UltraFeedback dataset (Cui et al., 2023) with Eq. 5, obtaining the EVOLVE *offline* model. For **online iterative training**, we construct new preference datasets through *Self-Refinement*: (1) we first sample 5K prompts from UltraFeedback, generate four responses per prompt with the EVOLVE *offline* model using iterative *Self-Refinement*, forming a dataset $\mathcal{D} = \{x^{(i)}, y_1^{(i)}, y_2^{(i)}, y_3^{(i)}, y_4^{(i)}\}$; (2) we then score these responses with the Skywork Reward Model (Liu et al., 2024), which is a compact yet high-performing model on the RewardBench leaderboard (Lambert et al., 2024b); (3) we construct a new preference dataset and train a new model, denoted EVOLVE *iter1*. Finally, we repeat the same procedure with another 10K prompts from UltraFeedback, producing the EVOLVE *iter2* model.

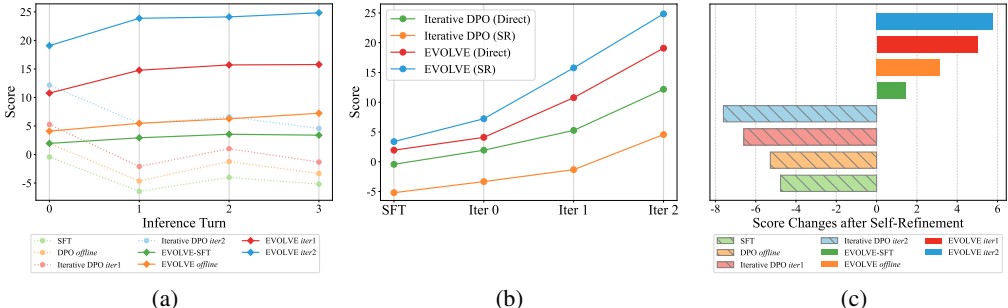

(a)                    (b)                    (c)

Figure 6: Evolution of *Self-Refinement* Capability through Iterative Training. (a) shows the evolution of *Self-Refinement* capability for each iteratively trained model during the inference stage. (b) depicts the performance progression of both the **Direct** and **SR** methods throughout iterative training. (c) demonstrates the performance improvement brought by **SR** compared to **Direct** across training iterations. Responses are generated from 256 UltraFeedback test set samples.

**Baselines.** . We first train the base model by applying standard SFT and DPO losses, following the same procedure and dataset as in our method, to obtain the SFT model and the DPO *offline* model. Based on these, we compare against several state-of-the-art baselines: (1) **Iterative DPO** (Snorkel, 2024; Xiong et al., 2023), an iterative preference optimization method; (2) **SynPO** (Dong et al., 2024), which leverages a refiner model to enhance performance; (3) **SRPO** (Choi et al., 2024), an offline preference optimization approach guided by an adversarial objective to improve response quality iteratively; and (4) **ScoRe** (Kumar et al., 2024), an online reinforcement learning approach for self-correction. Iterative DPO and SynPO are trained using DPO method, while SRPO is adapted for our iterative setting. Since ScoRe is designed for reasoning tasks, we integrate it into general tasks and perform RL training on the EVOLVE-SFT model. The specific implementation details of these methods are provided in Appendix B.

**Evaluation Benchmarks.** We evaluate our models on AlpacaEval 2 (Li et al., 2023) and Arena-Hard (Li et al., 2024), as well as on two cross-domain mathematical reasoning tasks: GSM8K (Cobbe et al., 2021) and MATH (Hendrycks et al., 2021). AlpacaEval 2 consists of 805 questions drawn from five datasets, while Arena-Hard comprises 500 well-defined technical questions. GSM8K contains elementary- and middle-school-level math problems, whereas MATH includes more challenging questions spanning diverse mathematical branches such as algebra, counting and probability, geometry, number theory, and calculus. For AlpacaEval 2, we report both the raw win rate (WR) and the length-controlled (LC) win rate (Dubois et al., 2024). For Arena-Hard, we measure the win rate relative to the baseline model. For GSM8K and MATH, we adopt the Step-DPO evaluation script (Lai et al., 2024). We assess all models under two settings: direct response generation (**Direct**) and responses refined through three rounds of *Self-Refinement* (**SR**).

## 4.2 BOOST IN MODEL PERFORMANCE WITH EVOLVE

**Consistent Gains in Inference and Iterative Training.** We begin by investigating the evolution of *Self-Refinement* capability during iterative training on the UltraFeedback test set, as shown in Fig. 6. EVOLVE demonstrates that *Self-Refinement* is not only activated but also progressively strengthened across iterations. As shown in Fig. 6(a) and Fig. 6(b), models trained with our framework continually improve during inference, while their refinement ability evolves steadily throughout iterative training. The training algorithm and *Self-Refinement* strategy complement each other to form a synergistic training loop that drives continual gains. Fig. 6(c) further shows that the gain from *Self-Refinement* over direct response generation increases with each iteration, confirming that EVOLVE enables models to gradually internalize and reinforce this cognitive mode.

**Superior Benchmark Performance over Baselines.** On AlpacaEval 2 and Arena-Hard (Tab. 1), EVOLVE achieves consistent improvements: it significantly boosts performance in the **Direct** setting and delivers a qualitative leap under the **SR** setting, demonstrating the benefits of progressive refinement. In contrast, preference-optimization methods built upon SFT often fail to enable effective *Self-Refinement*, even when the training objective explicitly includes it. Although ScoRe also shows strong refinement ability, RL-based approaches that rely on reward model scores frequently suffer from instability in general domains, largely due to reward hacking that undermines reliable optimization. These results confirm that EVOLVE provides more stable and comprehensive gains across benchmarks compared to existing methods.

Table 1: Performance comparison across AlpacaEval 2 and Arena-Hard benchmarks under the **Direct** and **SR** settings. Reported values are scores, with parentheses indicating gains over SFT baseline.

| Method | Direct | | | Self-Refinement (SR) | | |
|---|---|---|---|---|---|---|
| | AlpacaEval 2 | | Arena-Hard | AlpacaEval 2 | | Arena-Hard |
| | LC (%) | WR (%) | WR (%) | LC (%) | WR (%) | WR (%) |
| **Llama-3.1-8B Base** | | | | | | |
| SFT | 15.9 | 12.7 | 12.7 | 13.8 | 8.1 | 8.0 |
| +SRPO *offline* | 15.8 (-0.1) | 16.0 (+3.3) | 14.3 (+1.6) | 22.2 (+8.4) | 16.2 (+8.1) | 13.2 (+5.2) |
| +SRPO *iter*1 | 21.9 (+6.0) | 21.8 (+9.1) | 21.9 (+9.2) | 27.1 (+13.3) | 20.2 (+12.1) | 18.8 (+10.8) |
| +SRPO *iter*2 | 22.1 (+6.2) | 22.4 (+9.7) | 25.6 (+12.9) | 27.5 (+13.7) | 22.3 (+14.2) | 21.8 (+13.8) |
| +DPO *offline* | 17.9 (+2.0) | 16.7 (+4.0) | 16.5 (+3.8) | 18.3 (+4.5) | 12.6 (+4.5) | 12.6 (+4.6) |
| +SynPO *iter*1 | 23.5 (+7.6) | 23.1 (+10.4) | 21.6 (+8.9) | 22.5 (+8.7) | 15.1 (+7.0) | 14.0 (+6.0) |
| +SynPO *iter*2 | 23.1 (+7.2) | 18.2 (+5.5) | 19.6 (+6.9) | 20.6 (+6.8) | 13.8 (+5.7) | 16.8 (+8.8) |
| +Iterative DPO *iter*1 | 24.6 (+8.7) | 22.3 (+9.6) | 22.4 (+9.7) | 25.1 (+11.3) | 16.9 (+8.8) | 17.1 (+9.1) |
| +Iterative DPO *iter*2 | 34.1 (+18.2) | 33.5 (+20.8) | 29.6 (+16.9) | 34.5 (+20.7) | 27.8 (+19.7) | 23.7 (+15.7) |
| EVOLVE-SFT (Ours) | 15.9 (+0.0) | 15.5 (+2.8) | 16.5 (+3.8) | 20.0 (+6.2) | 18.6 (+10.5) | 18.0 (+10.0) |
| +ScoRe | 21.3 (+5.4) | 27.9 (+15.2) | 24.0 (+11.3) | 35.8 (+22.0) | 42.7 (+34.6) | 34.0 (+26.0) |
| +EVOLVE *offline* (Ours) | 19.1 (+3.2) | 18.6 (+5.9) | 17.4 (+4.7) | 28.8 (+15.0) | 27.1 (+19.0) | 23.5 (+15.5) |
| +EVOLVE *iter*1 (Ours) | 32.7 (+16.8) | 33.5 (+20.8) | 31.9 (+19.2) | 50.2 (+36.4) | 49.9 (+41.8) | 37.5 (+29.5) |
| +EVOLVE *iter*2 (Ours) | **45.0** (+29.1) | **46.8** (+34.1) | **38.0** (+25.3) | **62.3** (+48.5) | **63.3** (+55.2) | **50.3** (+42.3) |
| **Mistral-7B Base** | | | | | | |
| SFT | 17.0 | 15.8 | 11.7 | 9.2 | 5.6 | 4.0 |
| +DPO *offline* | 19.0 (+2.0) | 19.3 (+3.5) | 13.8 (+2.1) | 13.0 (+3.8) | 10.6 (+5.0) | 3.9 (-0.1) |
| +Iterative DPO *iter*1 | 29.7 (+12.7) | 33.5 (+17.7) | 22.8 (+11.1) | 15.3 (+6.1) | 17.6 (+12.0) | 13.7 (+9.7) |
| +Iterative DPO *iter*2 | **39.5** (+22.5) | **41.8** (+26.0) | 27.2 (+15.5) | 7.3 (-1.9) | 11.2 (+5.6) | 21.1 (+17.1) |
| EVOLVE-SFT (Ours) | 18.8 (+1.8) | 17.4 (+1.6) | 11.1 (-0.6) | 20.5 (+11.3) | 18.2 (+12.6) | 12.9 (+8.9) |
| +EVOLVE *offline* (Ours) | 16.2 (-0.8) | 16.7 (+0.9) | 13.3 (+1.6) | 23.3 (+14.1) | 22.7 (+17.1) | 15.6 (+11.6) |
| +EVOLVE *iter*1 (Ours) | 29.1 (+12.1) | 30.2 (+14.4) | 21.9 (+10.2) | 39.0 (+29.8) | 40.6 (+35.0) | 25.3 (+21.3) |
| +EVOLVE *iter*2 (Ours) | 39.3 (+22.3) | 40.2 (+24.4) | **27.7** (+16.0) | 46.4 (+37.2) | 49.7 (+44.1) | 32.8 (+28.8) |

## 4.3 Impact of Key Components on EVOLVE Performance

We analyze three critical factors in EVOLVE: (1) the role of the DPO loss in the training objective, (2) the importance of the *Self-Refinement* generation strategy for data collection, (3) the framework's dependence on the choice of reward model. The corresponding results are presented in Fig. 7, Fig. 8 and Appendix I.2.

**Impact of DPO Loss.** As shown in Fig. 7, omitting the DPO loss progressively degrades the performance of EVOLVE. With iterative training, the absence of DPO loss results in a slower improvement rate in both the **Direct** and **SR** settings. This highlights the crucial role of DPO loss in strengthening the model's ability to generate higher-quality responses directly.

**Impact of Self-Refinement Generation Strategy.** Fig. 8 shows that models trained on datasets collected via *Parallel Sampling* (PS) consistently underperform compared to those using the *Self-Refinement* generation strategy. This demonstrates that the *Self-Refinement* generation strategy is essential for constructing high-quality preference datasets. Further evidence is provided in Appendix L, where we show that *Self-Refinement* improves responses by enhancing logical coherence and producing clearer, more concise expressions.

**Impact of Reward Model Selection.** To assess the influence of reward models, we conducted ablation studies using the Skywork Reward Model (Liu et al., 2024) and ArmoRM (Wang et al., 2024). The results, shown in Tab. 10, demonstrate that models trained with ArmoRM exhibit performance comparable to those trained with the Skywork Reward Model. This indicates that our framework's effectiveness is independent of the reward model, and its ability to consistently enhance *Self-Refinement* capabilities remains robust across different reward model choices.

## 4.4 Generalization of EVOLVE to Out-of-Domain Reasoning Tasks

The results in Tab. 2 show that the *Self-Refinement* ability activated by EVOLVE generalizes effectively to reasoning tasks, yielding measurable performance gains on GSM8K and MATH.

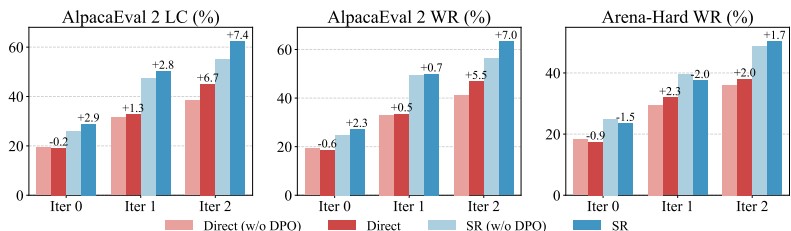

Figure 7: Impact analysis of DPO Loss on EVOLVE Performance.

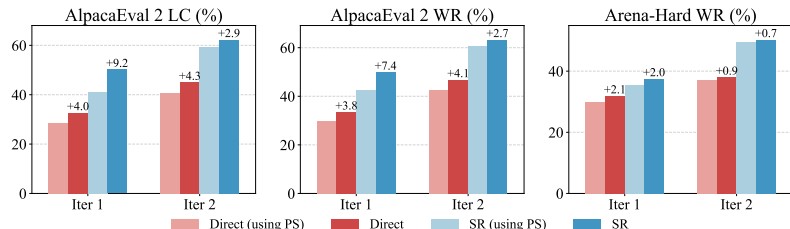

Figure 8: Effect of *Parallel Sampling* (PS) vs. *Self-Refinement* on EVOLVE Performance.

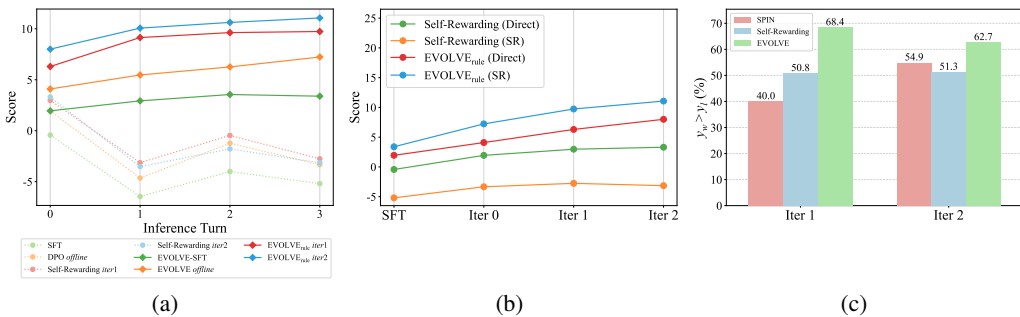

Figure 9: Model performance evaluation under the Self-Improvement setup: (a) shows performance variation with increasing inference turns. (b) depicts improvement during iterative training. (c) assesses the collected dataset quality using Skywork Reward Model.

This demonstrates that our method can transfer beyond its training domain and enhance reasoning ability without explicit exposure to mathematical data. For comparison, ScoRe also exhibits strong generalization on these tasks, which can be attributed to the inherent robust generalization properties of RL. However, as model accuracy on GSM8K and MATH increases, the improvements from *Self-Refinement* become less pronounced. We attribute this to the absence of domain-specific training, which prevents the model from mastering certain advanced problem-solving strategies. This observation motivates future work on incorporating domain-specific training for reasoning tasks, with the goal of further strengthening the reasoning capabilities of EVOLVE.

Table 2: Accuracy of Various Methods on GSM8K and MATH with Llama-3.1-8B.

| Method | GSM8K(%) | | MATH(%) | |
|---|---|---|---|---|
| | Direct | SR | Direct | SR |
| SFT | 60.9 | 62.3 | 18.2 | 29.5 |
| +SRPO *offline* | 66.7 | 63.9 | 36.2 | 34.5 |
| +SRPO *iter*1 | 67.2 | 66.6 | 38.5 | 37.1 |
| +SRPO *iter*2 | 65.0 | 68.7 | 41.5 | 39.3 |
| +DPO *offline* | 66.3 | 66.6 | 35.1 | 34.6 |
| +SynPO *iter*1 | 67.6 | 66.9 | 38.0 | 37.0 |
| +SynPO *iter*2 | 67.6 | 66.3 | 33.2 | 32.4 |
| +Iterative DPO *iter*1 | 68.5 | 67.9 | 36.6 | 35.6 |
| +Iterative DPO *iter*2 | 68.4 | 67.2 | 36.7 | 36.3 |
| EVOLVE-SFT | 64.3 | 70.9 | 32.8 | 45.5 |
| +ScoRe | 64.0 | 71.7 | 39.5 | **50.1** |
| +EVOLVE *offline* | 68.3 | 72.2 | 38.3 | 42.4 |
| +EVOLVE *iter*1 | 70.2 | 71.7 | 46.1 | 47.8 |
| +EVOLVE *iter*2 | **71.9** | **73.6** | **48.7** | **50.1** |

## 5 EXTENSION: THE POTENTIAL OF SELF-REFINEMENT FOR ACHIEVING MODEL SELF-IMPROVEMENT

Building on the discussion of activating and enhancing *Self-Refinement* capabilities, we now explore a compelling question: *Can the activated Self-Refinement ability of LLMs enable model intrinsic performance Self-Improvement?* We investigate this using our EVOLVE framework.

Table 3: Performance Analysis of *Self-Refinement* Potential for *Self-Improvement*. Results evaluated on AlpacaEval 2 (Li et al., 2023) and Arena-Hard (Li et al., 2024) under the **Direct** and **SR** settings. **Direct** denotes direct response generation, **SR** indicates three rounds of *Self-Refinement* on responses.

| Method | Direct | | | Self-Refinement (SR) | | |
|---|---|---|---|---|---|---|
| | AlpacaEval 2 | | Arena-Hard | AlpacaEval 2 | | Arena-Hard |
| | LC (%) | WR (%) | WR (%) | LC (%) | WR (%) | WR (%) |
| **Llama-3.1-8B Base** | | | | | | |
| SPIN *iter*1 | 15.5 | 12.8 | 11.1 | 17.2 | 12.4 | 10.2 |
| SPIN *iter*2 | 13.2 | 11.3 | 12.1 | 12.3 | 9.9 | 11.9 |
| Self-Rewarding *iter*1 | 19.3 | 17.2 | 14.2 | 19.3 | 12.6 | 11.6 |
| Self-Rewarding *iter*2 | 18.2 | 14.2 | 15.9 | 19.0 | 11.2 | 11.0 |
| EVOLVE$_{rule}$ *iter*1 | 23.9 | 24.5 | 22.0 | 37.4 | 35.7 | 31.1 |
| EVOLVE$_{rule}$ *iter*2 | **28.4** | **29.7** | **24.9** | **41.3** | **39.5** | **32.0** |
| **Mistral-7B Base** | | | | | | |
| SPIN *iter*1 | 14.8 | 13.3 | 7.2 | 8.6 | 7.5 | 3.3 |
| SPIN *iter*2 | 12.9 | 9.7 | 8.0 | 6.0 | 3.3 | 3.4 |
| Self-Rewarding *iter*1 | 21.0 | 18.0 | 13.2 | 13.0 | 9.5 | 4.7 |
| Self-Rewarding *iter*2 | 21.9 | 20.8 | 12.4 | 8.5 | 7.9 | 4.0 |
| EVOLVE$_{rule}$ *iter*1 | 19.7 | 21.0 | 14.2 | **28.2** | 28.7 | 17.9 |
| EVOLVE$_{rule}$ *iter*2 | **22.2** | **24.2** | **18.2** | 27.5 | **29.2** | **21.4** |

**Rules for Achieving Model Self-Improvement.** To objectively assess the model's *Self-Refinement* ability, **we employ a streamlined rule-based method for dataset filtering in this section**. Specifically, for a given problem $x$, the model generates a sequence of responses $\{y_1, y_2, y_3, y_4\}$. We then directly designate $y_1$ as the rejected response, while $y_4$ as the chosen response. These pairs are used to construct the preference dataset for subsequent preference training.

**Analysis Setup.** In this section, we focus on **online iterative training**. We select Self-Rewarding (Yuan et al., 2024) and SPIN (Chen et al., 2024) as baselines, which build upon the DPO *offline* model and are further trained using the same 5K-prompt dataset to produce the *iter*1 model, followed by a 10K-prompt dataset to yield the *iter*2 model, as described in Section 4.1. In contrast, our approach employs the **rule-based method** described above for data filtering, relying on the model's intrinsic *Self-Refinement* capability without external supervision. Starting from the EVOLVE *offline* model, we conduct two rounds of online training to obtain the EVOLVE *iter*1 and EVOLVE *iter*2 models.

**Evolution of Self-Refinement and Dataset Augmentation** We first investigate the evolution of *Self-Refinement* capability during iterative training for model *Self-Improvement* and analyze the quality of self-collected datasets, as shown in Fig. 9. Similar to experiments using reward models for data filtering, we observe that EVOLVE-trained models achieve synergistic improvements in both direct answering and *Self-Refinement* capabilities during iterative training. However, under unsupervised conditions, the rate of capability improvement slows. Additionally, we compare the quality of datasets collected by different methods, revealing that *Self-Refinement* enables superior performance gains by facilitating the collection of higher-quality datasets, as shown in Fig. 9(c).

**Challenges with Self-Rewarding and SPIN.** As shown in Table 3, Self-Rewarding and SPIN exhibit performance fluctuations and occasional declines, primarily due to the quality of their collected datasets. Self-Rewarding suffers from low dataset discriminability, resulting in stagnant performance. SPIN experiences further degradation due to even lower dataset quality, highlighting the challenges of achieving *Self-Improvement* in data-scarce domains.

**Summary.** *Self-Refinement* significantly enhances model performance by collecting higher-quality datasets without external supervision. However, experiments show it also introduces significant noise, highlighting the field's complexity and the need for further study. In Appendix F, we further explore the upper bounds of *Self-Improvement*, offering deeper insights into its potential.

# 6 CONCLUSION

We present EVOLVE, a novel framework that integrates iterative preference training with *Self-Refinement*-based inference. During training, EVOLVE enhances both direct question-answering performance and *Self-Refinement* capabilities. At inference time, it employs multi-step *Self-Refinement* to generate and refine outputs, creating preference data for subsequent training cycles. This synergistic loop leads to substantial improvements in LLM performance, surpassing GPT-4o on benchmarks such as AlpacaEval 2 and Arena-Hard. Furthermore, we investigate the potential of leveraging *Self-Refinement* to achieve *Self-Improvement* of the model's intrinsic abilities. While this work primarily focuses on general tasks, future research will explore extending this framework to more complex reasoning tasks and broader domains, including mathematics and coding.

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

## A  ADDITIONAL RELATED WORKS

**Reinforcement Learning from Human Feedback (RLHF).** RLHF has proven effective in aligning LLMs with human values (Christiano et al., 2017; Ouyang et al., 2022; Bai et al., 2022a; Song et al., 2023; Touvron et al., 2023). This approach uses human-annotated preference datasets to train a Reward Model, guiding LLM optimization through reinforcement learning. However, due to the high cost of human annotations, AI-generated feedback has been proposed to automate this process (Bai et al., 2022b; Lee et al., 2023). Additionally, to reduce training costs, Direct Preference Optimization (DPO) (Rafailov et al., 2023) bypasses the reward modeling process and directly aligns LLMs using preference datasets. However, the effectiveness of these methods heavily depends on the quality of the preference dataset, making the acquisition of high-quality preference data a critical challenge.

**In-Context Learning (ICL).** ICL has become a fundamental capability of LLMs, enabling them to perform tasks by conditioning on a few input examples without requiring parameter updates (Brown et al., 2020). Recent studies, such as OPRO (Yang et al., 2024), show that LLMs can leverage their ICL abilities to function as implicit optimizers, progressively improving performance on complex problems. LLMs can also act as in-context reinforcement learners, optimizing behavior via reward feedback (Monea et al., 2024). The SELF-REFINE (Madaan et al., 2024) is a special form of ICL. It significantly enhances model performance through the FEEDBACK and REFINE mechanisms, achieving remarkable results across multiple benchmarks. These findings indicate that integrating ICL with model training presents a compelling strategy for constructing self-optimizing frameworks.

## B  IMPLEMENTATION DETAILS OF BASELINES

**Iterative DPO** (Snorkel, 2024; Xiong et al., 2023): We conduct the training process based on the DPO *offline* model. Following the same online iterative training process as our method, we initially extract a 5K prompt data from the UltraFeedback dataset (Cui et al., 2023) dataset. For each prompt, we generate four responses in parallel. These responses are then scored and filtered using the Skywork Reward Model (Liu et al., 2024) to construct a preference dataset, which is used to train the Iterative DPO *iter1* model via DPO (Rafailov et al., 2023). Next, we extract a new 10K prompt data from the UltraFeedback dataset and repeat the above process to obtain the Iterative DPO *iter2* model.

**SynPO** (Dong et al., 2024): We utilize the 10K unused dataset from the UltraFeedback as Seed Data to train the Response Improver. In each iteration, the Response Improver is trained using SFT, taking a prompt $x$ and the response $y$ generated by the previous policy model $M_{t-1}$ as input, with the chosen response $y_w$ from the Seed Data as the target output. For the policy model, we first generate responses for the 5K prompt dataset from UltraFeedback using both the DPO *offline* model and the current policy model $M_{t-1}$, yielding $\{y_{\text{off}}^{(i)}\}$ and $\{y^{(i)}\}$, respectively. The policy model's responses $\{y^{(i)}\}$ are then refined by the Response Improver to produce $\{\hat{y}^{(i)}\}$. We then use the Skywork Reward Model to filter the $\{x^{(i)}, y_{\text{off}}^{(i)}, \hat{y}^{(i)}\}$ dataset, and train the DPO *offline* model to obtain the SynPO *iter1* model. This process is repeated with a new 10K prompt dataset to train the SynPO *iter2* model.

**SRPO** (Choi et al., 2024): We train the SRPO model starting from the SFT model, with two key modifications to the Iterative DPO process. First, we replace the training algorithm with SRPO across all stages. Second, as SRPO is an offline algorithm without a specified data collection process, we adopt our *Self-Refinement* approach for data collection. The offline data used, including its size, aligns with the training process of EVOLVE.

**ScoRe** (Kumar et al., 2024): Starting from the EVOLVE-SFT model, we use the first 45K prompts from the UltraFeedback dataset and apply the REINFORCE algorithm for online training, following ScoRe's two-stage training paradigm.

**Self-Rewarding** (Yuan et al., 2024): The training process of Self-Rewarding closely follows Iterative DPO, generating four responses per prompt through parallel sampling and training the model using DPO. The distinction lies in the construction of the preference dataset. Self-Rewarding employs *LLM-as-a-Judge* to score each prompt-response pair $(x, y)$, assigning a score to response $y$. In our experiments, the policy model itself serves as the *LLM-as-a-Judge*, evaluating the same dataset over 3 rounds and taking the average score as the final quality assessment. The highest-scoring response is selected as the chosen response and the lowest-scoring one as the rejected response. The *LLM-as-a-Judge* template used aligns with the Self-Rewarding paper. Through two iterative training

rounds on the collected 5K and 10K preference datasets, we obtained the Self-Rewarding *iter1* and Self-Rewarding *iter2* models.

**SPIN** (Chen et al., 2024): The training process of SPIN is similar to Iterative DPO, with the key difference lying in the data collection method. For each prompt $x$, SPIN uses model to generate a response $y$, which is directly treated as the rejected response. The chosen response $y_w$ from the preference dataset serves as the alignment target. Based on the DPO *offline* model, SPIN first collects preference data for 5K prompts from the UltraFeedback dataset and then conducts training to yield the SPIN *iter1* model. This process is then repeated with the SPIN *iter1* model and 10K prompt dataset from UltraFeedback, yielding the SPIN *iter2* model.

## C MATHEMATICAL DERIVATIONS

### C.1 THE OPTIMAL SOLUTION TO THE *Self-Refinement* OBJECTIVE IN THE PREFERENCE TRAINING PHASE

In this Appendix, we aim to derive the loss function corresponding to the following objective:

$$\max_{\pi} \ \mathbb{E}_{y_2 \sim \pi(\cdot|x,y_1,z)} \left[ p(y_2 \succ y_1|x) - \beta D_{\mathrm{KL}}(\pi||\pi_{\mathrm{ref}}|x,y_1,z) \right]. \tag{6}$$

First, we can obtain the optimal solution of the objective:

$$\max_{\pi} \ \mathbb{E}_{y_2 \sim \pi(\cdot|x,y_1,z)} \left[ p(y_2 \succ y_1|x) - \beta D_{\mathrm{KL}}(\pi||\pi_{\mathrm{ref}}|x,y_1,z) \right] \tag{7}$$

$$= \max_{\pi} \ \mathbb{E}_{y_2 \sim \pi(\cdot|x,y_1,z)} \left[ p(y_2 \succ y_1|x) - \beta \log \frac{\pi(y_2|x,y_1,z)}{\pi_{\mathrm{ref}}(y_2|x,y_1,z)} \right] \tag{8}$$

$$= \max_{\pi} \ \beta \mathbb{E}_{y_2 \sim \pi(\cdot|x,y_1,z)} \left[ - \log \frac{\pi(y_2|x,y_1,z)}{\pi_{\mathrm{ref}}(y_2|x,y_1,z) \exp\left(\frac{p(y_2 \succ y_1|x)}{\beta}\right)} \right] \tag{9}$$

$$= \max_{\pi} \ -\beta \mathbb{E}_{y_2 \sim \pi(\cdot|x,y_1,z)} \left[ \log \frac{\pi(y_2|x,y_1,z) Z(x,y_1,z)}{\pi_{\mathrm{ref}}(y_2|x,y_1,z) \exp\left(\frac{p(y_2 \succ y_1|x)}{\beta}\right)} \right] + \beta \log Z(x,y_1,z) \tag{10}$$

$$= \max_{\pi} \ -\beta D_{\mathrm{KL}} \left( \pi(y_2|x,y_1,z) \middle\| \frac{\pi_{\mathrm{ref}}(y_2|x,y_1,z) \exp\left(\frac{p(y_2 \succ y_1|x)}{\beta}\right)}{Z(x,y_1,z)} \right) + \beta \log Z(x,y_1,z) \tag{11}$$

where $Z(x,y_1,z)$ is the partition function. Considering the non-negativity of the KL divergence, the optimal solution is:

$$\pi^*(y_2|x,y_1,z) = \frac{\pi_{\mathrm{ref}}(y_2|x,y_1,z) \exp\left(\frac{p(y_2 \succ y_1|x)}{\beta}\right)}{Z(x,y_1,z)}. \tag{12}$$

Noting that $p(y_1 \succ y_1|x) = 1/2$, we derive the following expression:

$$\pi^*(y_1|x,y_1,z) = \frac{\pi_{\mathrm{ref}}(y_1|x,y_1,z) \exp\left(\frac{1}{2\beta}\right)}{Z(x,y_1,z)}. \tag{13}$$

Dividing Eq.12 by Eq.13 yields

$$\frac{\pi^*(y_2|x,y_1,z)}{\pi^*(y_1|x,y_1,z)} = \frac{\pi_{\mathrm{ref}}(y_2|x,y_1,z)}{\pi_{\mathrm{ref}}(y_1|x,y_1,z)} \exp\left(\frac{p(y_2 \succ y_1|x)}{\beta} - \frac{1}{2\beta}\right). \tag{14}$$

Therefore, we have

$$\log \left( \frac{\pi^*(y_2|x,y_1,z) \pi_{\mathrm{ref}}(y_1|x,y_1,z)}{\pi^*(y_1|x,y_1,z) \pi_{\mathrm{ref}}(y_2|x,y_1,z)} \right) = \frac{p(y_2 \succ y_1|x)}{\beta} - \frac{1}{2\beta}. \tag{15}$$

By adopting the mean squared error as the loss function and parametrizing the policy model as $\pi_\theta$, we finally obtain:

$$\mathcal{L}(\pi_\theta; \pi_{\text{ref}}) = \mathbb{E}_{(x,y_1,y_2)\sim\rho}\left[\log\left(\frac{\pi_\theta(y_2|x,y_1,z)\pi_{\text{ref}}(y_1|x,y_1,z)}{\pi_\theta(y_1|x,y_1,z)\pi_{\text{ref}}(y_2|x,y_1,z)}\right) - \left(\frac{p(y_2 \succ y_1|x)}{\beta} - \frac{1}{2\beta}\right)\right]^2, \tag{16}$$

where $\rho$ represents the true distribution.

## D  IMPLEMENTATION DETAILS

### D.1  SELF-REFINEMENT TEMPLATE

The *Self-Refinement* template used in this paper is as follows:

---

**Self-Refinement Template**

Below is a QUESTION from a user and an EXAMPLE RESPONSE.
Please provide a more helpful RESPONSE, improving the EXAMPLE RESPONSE by making the content even clearer, more accurate, and concise. Focus on addressing the human's QUESTION without including irrelevant sentences.
Your RESPONSE should not only be well-written, logical, and easy-to-follow, but also demonstrate expert-level insight, engaging the reader with the most relevant information.

QUESTION:
{Question}

EXAMPLE RESPONSE:
{Example_Response}

Now, refine and improve the RESPONSE further. You can consider two approaches:
1. REFINEMENT: If the EXAMPLE RESPONSE is sufficient and addresses most of the QUESTION's concerns, enhance clarity, accuracy, or conciseness as needed.
2. NEW RESPONSE: If the EXAMPLE RESPONSE lacks clarity or relevance to the QUESTION, craft a more effective RESPONSE that thoroughly resolves the QUESTION.

Do not include analysis-just give the improved RESPONSE.

RESPONSE:

---

### D.2  TRAINING DETAILS

In the SFT phase, we set the learning rate to $5 \times 10^{-6}$, with a batch size of 128 and a maximum sequence length of 1024. We employed a cosine learning rate schedule with 3% warm-up steps for 1 epoch and used the AdamW optimizer.

In the preference training phase, the learning rate was reduced to $1 \times 10^{-6}$. Additionally, we set $\alpha = 0.8$. For training with the Reward Model Scoring dataset filtering mechanism, we set $\beta = 0.01$, while for the Self-Improvement Rule-Based Selection mechanism, $\beta = 0.05$. The higher value of $\beta$ in the Self-Improvement Rule-Based Selection process is due to the presence of noisy data in the filtered preference dataset, which requires stronger regularization.

### D.3  INFERENCE DETAILS

During the iterative training and dataset collection process, we employed a sampling decoding strategy with a temperature of 0.7 for direct response generation and *Self-Refinement*. For AlpacaEval 2, we set the temperature to 0.9 for generation, while for MT-Bench and Arena-Hard, we followed the official decoding configuration. For GSM8K and MATH, we utilized a greedy decoding strategy.

## D.4 EVALUATION BENCHMARKS DETAILS

AlpacaEval 2 (Li et al., 2023) consists of 805 questions from 5 datasets, MT-Bench (Zheng et al., 2023) covers 80 questions across 8 categories in a multi-turn dialogue format, and Arena-Hard (Li et al., 2024) is an enhanced version of MT-Bench with 500 well-defined technical questions. GSM8K (Cobbe et al., 2021) includes elementary and middle school-level math problems, while MATH (Hendrycks et al., 2021) contains more complex questions, spanning various mathematical branches such as algebra, counting and probability, geometry, number theory, and calculus. We evaluate the entire GSM8K test set, which contains 1319 math problems, and the first 1024 samples of the MATH test set. For each benchmark, we report scores according to their respective evaluation protocols.

## D.5 EXPERIMENT RESULTS ON MT-BENCH

For MT-Bench (Zheng et al., 2023), we report the average score using GPT-4 and GPT-4-Preview-1106 as judges.

Table 4: Results on and MT-Bench (Zheng et al., 2023).

| Method | MT-Bench | |
|---|---|---|
| | GPT-4 Turbo | GPT-4 |
| SFT | 6.4 | 6.9 |
| +DPO *offline* | 6.9 | 7.4 |
| +Self-Rewarding *iter*1 | 6.9 | 7.5 |
| +Self-Rewarding *iter*2 | 6.7 | 7.4 |
| +Iterative DPO *iter*1 | 6.9 | 7.6 |
| +Iterative DPO *iter*2 | 7.1 | 7.8 |
| EVOLVE-SFT | 6.4 | 7.0 |
| +EVOLVE *offline* | 7.0 | 7.6 |
| +EVOLVE$_{rule}$ *iter*1 | 7.1 | 7.5 |
| +EVOLVE$_{rule}$ *iter*2 | 7.1 | 7.7 |
| +EVOLVE *iter*1 | 7.3 | 7.7 |
| +EVOLVE *iter*2 | **7.7** | **8.1** |

## D.6 COMPUTATIONAL OVERHEAD OF TRAINING AND INFERENCE

We compare the computational overhead between Iterative DPO and EVOLVE in the two-round online iterative training, with the results presented as follows:

Table 5: Computational overhead during the training and inference stage.

| Method | Iter1 | | Iter2 | |
|---|---|---|---|---|
| | Dataset Collection (5k) | Training (5k) | Dataset Collection (10k) | Training (10k) |
| Iterative DPO | 4h | 25min | 8h | 50min |
| EVOLVE | 5.5h | 50min | 11h | 100min |

To ensure a fair comparison, we introduce a normalization parameter $\alpha$ into our loss function, such that the overall weight of the loss function during updates remains equal to 1.

## D.7 ABLATION STUDY ON PARAMETER $\alpha$

Fig. 10 shows the experimental results with different values of $\alpha$ on Llama-3.1-8B Base. Based on the results which demonstrate the *Self-Refinement* capability performs optimally at $\alpha = 0.8$, we selected $\alpha = 0.8$ in this work.

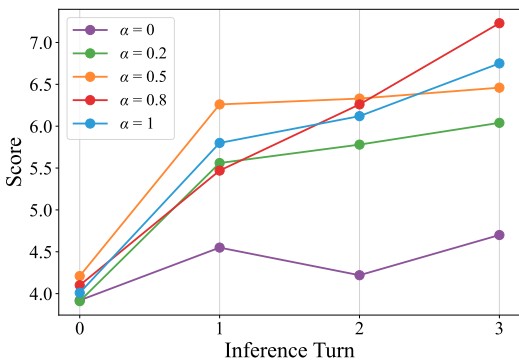

Figure 10: Experimental results with different values of $\alpha$ on Llama-3.1-8B Base.

## E   ALTERNATIVE REFINEMENT LOSS

### E.1   THE DERIVATION OF THE REFINEMENT LOSS FUNCTION FROM THE BRADLEY-TERRY MODEL PERSPECTIVE

Alternatively, we can enhance the model's *Self-Refinement* capability by leveraging insights from the Bradley-Terry (BT) model theory. We define the objective function as follows:

$$\max_{\pi} \ \mathbb{E}_{y_2 \sim \pi(\cdot|x,y_1,z)}\left[r(y_2|x,y_1,z) - \beta D_{\mathrm{KL}}(\pi||\pi_{\mathrm{ref}}|x,y_1,z)\right] \tag{17}$$

The solution process is analogous to that of Appendix C.1, allowing us to obtain the optimal solution:

$$\pi^*(y_2|x,y_1,z) = \frac{\pi_{\mathrm{ref}}(y_2|x,y_1,z)\exp\left(\frac{r(y_2|x,y_1,z)}{\beta}\right)}{Z(x,y_1,z)}, \tag{18}$$

where $Z(x,y_1,z)$ is the partition function. Reorganizing the above equation, we obtain:

$$r(y_2|x,y_1,z) = \beta\log\frac{\pi(y_2|x,y_1,z)}{\pi_{\mathrm{ref}}(y_2|x,y_1,z)} + \beta\log Z(x,y_1,z) \tag{19}$$

The standard expression of the BT model is:

$$p^*_{\mathrm{BT}}(y_2 \succ y_1|x) = \sigma(r^*(y_2|x) - r^*(y_1|x)) \tag{20}$$

Here, to enhance the *Self-Refinement* capability of the language model, we make a slight modification. Given the problem input $x$ for the BT model, we also provide an arbitrary response $y_{\mathrm{opt}}$ along with a refinement template $z$, which serves as guidance for the model to generate better responses:

$$p^*_{\mathrm{BT}}(y_2 \succ y_1|x,y_{\mathrm{opt}},z) = \sigma(r^*(y_2|x,y_{\mathrm{opt}},z) - r^*(y_1|x,y_{\mathrm{opt}},z)). \tag{21}$$

Then we define the refinement preference function:

$$p^*_{\mathrm{BT\_refine}}(y_2 \succ y_1|x,z) = p^*_{\mathrm{BT}}(y_2 \succ y_1|x,y_1,z)p^*_{\mathrm{BT}}(y_2 \succ y_1|x,y_2,z) \tag{22}$$

$$= \sigma(r^*(y_2|x,y_1,z) - r^*(y_1|x,y_1,z))\sigma(r^*(y_2|x,y_2,z) - r^*(y_1|x,y_2,z)) \tag{23}$$

$$= \sigma\left(\beta\log\frac{\pi^*(y_2|x,y_1,z)}{\pi_{\mathrm{ref}}(y_2|x,y_1,z)} - \beta\log\frac{\pi^*(y_1|x,y_1,z)}{\pi_{\mathrm{ref}}(y_1|x,y_1,z)}\right)$$
$$\times \sigma\left(\beta\log\frac{\pi^*(y_2|x,y_2,z)}{\pi_{\mathrm{ref}}(y_2|x,y_2,z)} - \beta\log\frac{\pi^*(y_1|x,y_2,z)}{\pi_{\mathrm{ref}}(y_1|x,y_2,z)}\right) \tag{24}$$

Assuming access to a well-curated preference dataset $\mathcal{D} = \{(x^{(i)}, y_w^{(i)}, y_l^{(i)})\}_{i=1}^N$, we aim to leverage this dataset to activate the *Self-Refinement* capability of language models, thereby gradually steering

the models toward generating better responses during the inference phase. To achieve this, we parametrize the policy model $\pi_\theta$ and estimate its parameters through maximum likelihood estimation. By treating the problem as a binary classification task, we have the negative log-likelihood loss:

$$
\begin{aligned}
\mathcal{L}_{\mathrm{BT\_SR}}(\pi) = & -\mathop{\mathbb{E}}_{(x,y_w,y_l)\sim\mathcal{D}}\left[\log\sigma\left(\beta\log\frac{\pi_\theta(y_w|x,y_l,z)}{\pi_{\mathrm{ref}}(y_w|x,y_l,z)}-\beta\log\frac{\pi_\theta(y_l|x,y_l,z)}{\pi_{\mathrm{ref}}(y_l|x,y_l,z)}\right)\right] \\
& -\mathop{\mathbb{E}}_{(x,y_w,y_l)\sim\mathcal{D}}\left[\log\sigma\left(\beta\log\frac{\pi_\theta(y_l|x,y_w,z)}{\pi_{\mathrm{ref}}(y_l|x,y_w,z)}-\beta\log\frac{\pi_\theta(y_w|x,y_w,z)}{\pi_{\mathrm{ref}}(y_w|x,y_w,z)}\right)\right]
\end{aligned}
\tag{25}
$$

Finally, we integrate the *Self-Refinement* loss with the DPO loss derived from the BT model perspective to obtain the EVOLVE loss function from the BT model viewpoint:

$$
\mathcal{L}_{\mathrm{BT\_EVOLVE}}(\pi_\theta;\pi_{\mathrm{ref}}) = \alpha\mathcal{L}_{\mathrm{BT\_SR}}(\pi_\theta;\pi_{\mathrm{ref}}) + (1-\alpha)\mathcal{L}_{\mathrm{DPO}}(\pi_\theta;\pi_{\mathrm{ref}}).
\tag{26}
$$

### E.2 Training Details

During the SFT phase, BT_EVOLVE follows the same procedure as EVOLVE. The primary distinction between BT_EVOLVE and EVOLVE arises in the preference training phase, where we set $\beta = 0.05$ in BT_EVOLVE.

### E.3 Experiment Results

We compared BT_EVOLVE and EVOLVE across various benchmarks, with experimental results presented in Table 6. Both BT_EVOLVE and EVOLVE demonstrated nearly identical performance across all benchmarks, underscoring that the strength of our approach lies not in the algorithm itself, but in the foundational principles it embodies. Specifically, it is the concept of refinement that drives the effectiveness of our method and framework, enabling them to deliver impressive results.

Table 6: Comparison of experimental results between BT_EVOLVE and EVOLVE on AlpacaEval 2 (Li et al., 2023), Arena-Hard (Li et al., 2024), and MT-Bench (Zheng et al., 2023) under the **Direct** and **SR** settings. LC and WR represent length-controlled win rate and raw win rate, respectively.

| Method | Direct | | | | | Self-Refinement (SR) | | |
| --- | --- | --- | --- | --- | --- | --- | --- | --- |
| | AlpacaEval 2 | | Arena-Hard | MT-Bench | | AlpacaEval 2 | | Arena-Hard |
| | LC (%) | WR (%) | WR (%) | GPT-4 Turbo | GPT-4 | LC (%) | WR (%) | WR (%) |
| BT_EVOLVE *offline* | 19.8 | 19.3 | 20.0 | 7.0 | 7.5 | 27.8 | 25.7 | 24.8 |
| BT_EVOLVE *iter*1 | 31.9 | 34.5 | 31.1 | 7.1 | 7.5 | 50.6 | 51.8 | 41.0 |
| BT_EVOLVE *iter*2 | **45.2** | **47.7** | **39.5** | 7.4 | 7.7 | **66.2** | **66.6** | 49.9 |
| EVOLVE *offline* | 19.1 | 18.6 | 17.4 | 7.0 | 7.6 | 28.8 | 27.1 | 23.5 |
| EVOLVE *iter*1 | 32.7 | 33.5 | 31.9 | 7.3 | 7.7 | 50.2 | 49.9 | 37.5 |
| EVOLVE *iter*2 | 45.0 | 46.8 | 38.0 | **7.7** | **8.1** | 62.3 | 63.3 | **50.3** |

Table 7: Accuracy Comparison of BT_EVOLVE and EVOLVE on GSM8K (Cobbe et al., 2021) and MATH (Hendrycks et al., 2021) tasks using **Direct** and **SR** generation strategies.

| Method | GSM8K(%) | | MATH(%) | |
| --- | --- | --- | --- | --- |
| | Direct | SR | Direct | SR |
| BT_EVOLVE *offline* | 67.6 | 71.7 | 37.7 | 44.1 |
| BT_EVOLVE *iter*1 | 70.3 | 73.2 | 46.5 | 47.4 |
| BT_EVOLVE *iter*2 | 70.1 | 71.6 | **50.2** | **52.0** |
| EVOLVE *offline* | 68.3 | 72.2 | 38.3 | 42.4 |
| EVOLVE *iter*1 | 70.2 | 71.7 | 46.1 | 47.8 |
| EVOLVE *iter*2 | **71.9** | **73.6** | 48.7 | 50.1 |

# F  FURTHER EXPLORATION OF SELF-IMPROVEMENT WITH MORE ITERATIONS

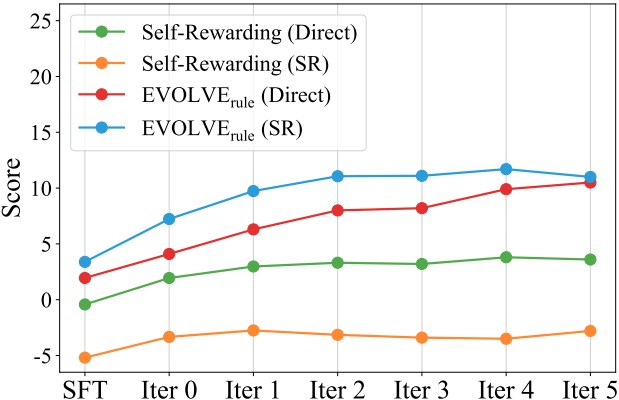

Figure 11: Investigation of the upper bound of the training iterations. We divided the last 15K samples from the UltraFeedback dataset into 3 subsets (5K each) and performed three additional training iterations. The evaluations are performed on the UltraFeedback test dataset.

As shown in Fig. 11, EVOLVE$_{rule}$'s performance of *Self-Refinement* (**SR**) has essentially reached its upper bound, exhibiting stabilization with minor fluctuation within a certain range. Meanwhile, the performance of Direct Response Generation (**Direct**) continues to improve. Based on this trend, the performance of **Direct** is expected to approach that of SR, achieving little or almost no gap. Besides, the Self-Rewarding demonstrates performance convergence by Iter 3 according to the trend.

# G  COMPARISON WITH OPEN-SOURCE MODELS

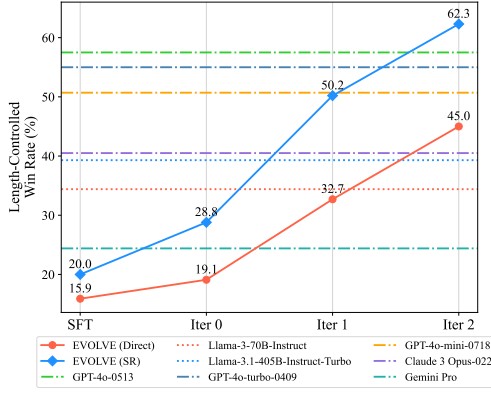

Figure 12: Length-controlled win rate on AlpacaEval 2 improves with EVOLVE iterations, surpassing GPT-4 level for the base versions of Llama-3.1-8B when utilizing the *Self-Refinement* strategy.

| Model | Size | LC(%) | WR(%) |
|---|---|---|---|
| **Llama-3.1-8B-Base-EVOLVE** (*iter*2, SR) | 8B | 62.3 | 63.3 |
| GPT-4o (05/13) | - | 57.5 | 51.3 |
| GPT-4-turbo (04/09) | - | 55.0 | 46.1 |
| GPT-4o-mini (07/18) | - | 50.7 | 44.7 |
| **Llama-3.1-8B-Base-EVOLVE** (*iter*1, SR) | 8B | 50.2 | 49.9 |
| GPT-4_1106_preview | - | 50.0 | 50.0 |
| **Llama-3.1-8B-Base-EVOLVE** (*iter*2, Direct) | 8B | 45.0 | 46.8 |
| Claude 3 Opus (02/29) | - | 40.5 | 29.1 |
| Llama-3.1-405B-Instruct-Turbo | 405B | 39.3 | 39.1 |
| Qwen2-72B-Instruct | 72B | 38.1 | 29.9 |
| Llama-3-70B-Instruct | 70B | 34.4 | 33.2 |
| **Llama-3.1-8B-Base-EVOLVE** (*iter*1, Direct) | 8B | 32.7 | 33.5 |
| Mistral Large (24/02) | 123B | 32.7 | 21.4 |
| Gemini Pro | - | 24.4 | 18.2 |
| Llama-3.1-8B-Instruct | 8B | 20.9 | 21.8 |

Table 8: Results on AlpacaEval 2 leaderboard. LC and WR represent length-controlled and raw win rate, respectively. "Direct" refers to the direct response generation strategy, while "SR" denotes the *Self-Refinement* generation strategy.

# H QUALITY EVALUATION OF DATA COLLECTED UNDER THE SELF-IMPROVEMENT SETTING

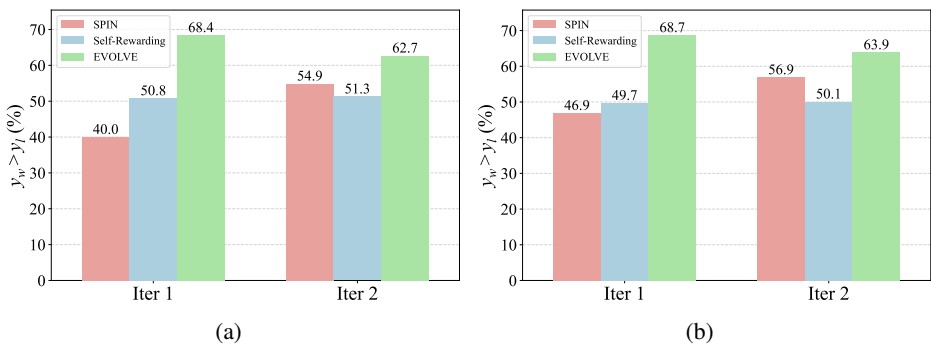

Figure 13: Quality Evaluation of the Dataset Collected under the Self-Improvement Setting. We evaluate the dataset collected without supervised signals using two reward models: Skywork Reward Model (Liu et al., 2024) (Figure a) and ArmoRM (Wang et al., 2024) (Figure b). Both reward models are 8B Parameter Scale and have demonstrated outstanding performance on the RewardBench leaderboard (Lambert et al., 2024b). The y-axis indicates the percentage of samples in the dataset for which the chosen response achieves a higher reward score compared to the rejected response. By leveraging different reward models for scoring, we aim to minimize potential biases in the evaluation results. The assessments from both reward models consistently demonstrate the effectiveness of the Self-Refinement generation strategy in enhancing the dataset quality.

# I DETAILS OF ABLATION STUDIES

## I.1 DPO LOSS AND GENERATION STRATEGY

The specific experimental results for Fig. 7 and Fig. 8 are presented in Tab. 9 below.

Table 9: Ablation study on AlpacaEval 2 and Arena-Hard. LC and WR represent length-controlled win rate and raw win rate, respectively.

| Method | Direct | | | Self-Refinement (SR) | | |
|---|---|---|---|---|---|---|
| | AlpacaEval 2 | | Arena-Hard | AlpacaEval 2 | | Arena-Hard |
| | LC (%) | WR (%) | WR (%) | LC (%) | WR (%) | WR (%) |
| EVOLVE *iter*1 using PS | 28.7 | 29.7 | 29.8 | 41.0 | 42.5 | 35.5 |
| EVOLVE *iter*2 using PS | 40.7 | 42.7 | 37.1 | 59.4 | 60.6 | 49.6 |
| EVOLVE *offline* w/o DPO | 19.3 | 19.2 | 18.3 | 25.9 | 24.8 | 25.0 |
| EVOLVE *iter*1 w/o DPO | 31.4 | 33.0 | 29.6 | 47.4 | 49.2 | 39.5 |
| EVOLVE *iter*2 w/o DPO | 38.3 | 41.3 | 36.0 | 54.9 | 56.3 | 48.6 |
| EVOLVE *offline* | 19.1 | 18.6 | 17.4 | 28.8 | 27.1 | 23.5 |
| EVOLVE *iter*1 | 32.7 | 33.5 | 31.9 | 50.2 | 49.9 | 37.5 |
| EVOLVE *iter*2 | **45.0** | **46.8** | **38.0** | **62.3** | **63.3** | **50.3** |

## I.2 IMPACT OF DIFFERENT REWARD MODELS

We also conducted ablation studies on the influence of different reward models adopted during the entire iterative training process. The results are presented in Tab. 10. It can be seen that the performance of the models trained with ArmoRM is generally consistent with that of the models trained with the Skywork Reward Model reported in this paper. This suggests that the effectiveness of our framework is not dependent on the reward model, and its ability to continually enhance the Self-Refinement capability is robust to this choice.

Table 10: Results on AlpacaEval 2 (Li et al., 2023) and Arena-Hard (Li et al., 2024) under the **Direct** and **SR** settings, when using **Skywork Reward Model** and **ArmoRM** as the reward model to retrain the entire iterative process based on the EVOLVE *offline* model, respectively.

| Method | Direct | | | Self-Refinement (SR) | | |
|---|---|---|---|---|---|---|
| | AlpacaEval 2 | | Arena-Hard | AlpacaEval 2 | | Arena-Hard |
| | LC (%) | WR (%) | WR (%) | LC (%) | WR (%) | WR (%) |
| SFT | 15.9 | 12.7 | 12.7 | 13.8 | 8.1 | 8.0 |
| EVOLVE-SFT | 15.9 | 15.5 | 16.5 | 20.0 | 18.6 | 18.0 |
| EVOLVE *offline* | 19.1 | 18.6 | 17.4 | 28.8 | 27.1 | 23.5 |
| **Skywork Reward Model** | | | | | | |
| EVOLVE *iter*1 | 32.7 | 33.5 | 31.9 | 50.2 | 49.9 | 37.5 |
| EVOLVE *iter*2 | 45.0 | **46.8** | 38.0 | 62.3 | **63.3** | 50.3 |
| **ArmoRM** | | | | | | |
| EVOLVE *iter*1 | 34.1 | 28.8 | 29.3 | 48.7 | 35.7 | 39.1 |
| EVOLVE *iter*2 | **45.4** | 42.5 | **40.6** | **63.8** | 58.1 | **56.4** |

## J   EVALUATION OF SELF-REFINEMENT CAPABILITIES IN OPEN-SOURCE MODELS

In this section, we evaluate the *Self-Refinement* capabilities of several prominent open-source models. To provide a comprehensive assessment, we examine the performance of the Llama-3.1-8B-Instruct, Qwen2.5-7B-Instruct, and Gemma-2-9B-Instruct models across three different refinement templates. The templates employed in this evaluation are as follows:

---

**Direct Refinement Template**

Below is a QUESTION from a user and an EXAMPLE RESPONSE.
Please provide a more helpful RESPONSE, improving the EXAMPLE RESPONSE by making the content even clearer, more accurate, and concise. Focus on addressing the human's QUESTION without including irrelevant sentences.
Your RESPONSE should not only be well-written, logical, and easy-to-follow, but also demonstrate expert-level insight, engaging the reader with the most relevant information.

QUESTION:
{Question}

EXAMPLE RESPONSE:
{Example_Response}

Now, refine and improve the RESPONSE further. You can consider two approaches:
1. REFINEMENT: If the EXAMPLE RESPONSE is sufficient and addresses most of the QUESTION's concerns, enhance clarity, accuracy, or conciseness as needed.
2. NEW RESPONSE: If the EXAMPLE RESPONSE lacks clarity or relevance to the QUESTION, craft a more effective RESPONSE that thoroughly resolves the QUESTION.

Do not include analysis-just give the improved RESPONSE.

RESPONSE:

---

---

### Analysis-Guided Refinement Template

Below is a QUESTION from a user and an EXAMPLE RESPONSE.
Please provide a more helpful RESPONSE, improving the EXAMPLE RESPONSE by making the content even clearer, more accurate, and concise. Focus on addressing the human's QUESTION without including irrelevant sentences.
Your RESPONSE should not only be well-written, logical, and easy-to-follow, but also demonstrate expert-level insight, engaging the reader with the most relevant information.

QUESTION:
{Question}

EXAMPLE RESPONSE:
{Example_Response}

Now, refine and improve the RESPONSE further. You can consider two approaches:
1. REFINEMENT: If the EXAMPLE RESPONSE is sufficient and addresses most of the QUESTION's concerns, enhance clarity, accuracy, or conciseness as needed.
2. NEW RESPONSE: If the EXAMPLE RESPONSE lacks clarity or relevance to the QUESTION, craft a more effective RESPONSE that thoroughly resolves the QUESTION.

Format your answer as follows:
ANALYSIS: <Analyze the strengths and shortcomings of the EXAMPLE RESPONSE>
RESPONSE: <Provide an improved response>

---

### Minimal-Prompt Refinement Template

Below is a QUESTION from a user and an EXAMPLE RESPONSE.
Please provide a better RESPONSE.

QUESTION:
{Question}

EXAMPLE RESPONSE:
{Example_Response}

RESPONSE:

---

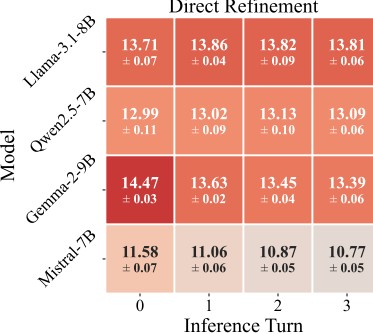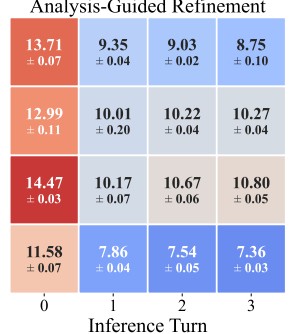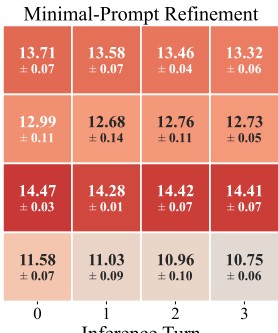

Figure 14: Evaluation of *Self-Refinement* Capability Across Various Models. We use three refinement templates to minimize prompt bias. The x-axis denotes the inference iteration number. For each turn, responses are generated from 256 UltraFeedback test set samples, using the original prompt and the prior turn's output. These are then scored by the ArmoRM (Wang et al., 2024). To eliminate potential randomness, the reported values are the mean score of three independent runs with different random seeds. For better visualization, ArmoRM scores are scaled by a factor of 100 due to their originally narrow value range; higher scores indicate better quality.

To ensure robust evaluation and mitigate biases from a single reward model, we employed two distinct reward models—Skywork Reward Model (Liu et al., 2024) and ArmoRM (Wang et al., 2024)—to score the results, as shown in Fig. 1 (Skywork Reward Model) and Fig. 14 (ArmoRM). Our experiments demonstrate that, despite their widespread adoption, current open-source models often struggle to effectively refine their responses, which can even lead to a degradation in performance.

## K    ANALYSIS OF VARIOUS DYNAMIC GENERATION STRATEDIES

**Prompt Template of Chain of Self-Refinement Generation Strategy**:

---

### Chain of Self-Refinement Template

Below is a QUESTION from a user and an EXAMPLE RESPONSE.
Please provide a more helpful RESPONSE, improving the EXAMPLE RESPONSE by making the content even clearer, more accurate, and concise. Focus on addressing the human's QUESTION without including irrelevant sentences.
Your RESPONSE should not only be well-written, logical, and easy-to-follow, but also demonstrate expert-level insight, engaging the reader with the most relevant information.

QUESTION:
{Question}

EXAMPLE RESPONSE:
{Example_Response}

Now, refine and improve the RESPONSE further. You can consider two approaches:
1. REFINEMENT: If the EXAMPLE RESPONSE is sufficient and addresses most of the QUESTION's concerns, enhance clarity, accuracy, or conciseness as needed.
2. NEW RESPONSE: If the EXAMPLE RESPONSE lacks clarity or relevance to the QUESTION, craft a more effective RESPONSE that thoroughly resolves the QUESTION.

Do not include analysis-just give the improved RESPONSE.

RESPONSE:

---

**Prompt Template of Few-Shot Self-Refinement Generation Strategy**:

---

### Few-Shot Self-Refinement Template

Below is a QUESTION from a user and several EXAMPLE RESPONSES, ordered in a *Self-Refinement* sequence.
Please provide a more helpful RESPONSE, improving the previous EXAMPLE RESPONSES by making the content even clearer, more accurate, and concise. Focus on addressing the human's QUESTION without including irrelevant sentences.
Your RESPONSE should not only be well-written, logical, and easy-to-follow, but also demonstrate expert-level insight, engaging the reader with the most relevant information.

QUESTION:
{Question}

EXAMPLE RESPONSES (in *Self-Refinement* order):
{Example_Responses}

Now, refine and improve the RESPONSE further. You can consider two approaches:
1. REFINEMENT: If the EXAMPLE RESPONSES are sufficient and addresses most of the QUES-TION's concerns, enhance clarity, accuracy, or conciseness as needed.
2. NEW RESPONSE: If the EXAMPLE RESPONSES lack clarity or relevance to the QUESTION, craft a more effective RESPONSE that thoroughly resolves the QUESTION.

Do not include analysis-just give the improved RESPONSE.

RESPONSE:

---

**Self-Refinement with Self-Evaluation Generation Strategy**: The Self-Refinement with Self-Evaluation generation strategy adopts the same Self-Refinement template as the Chain of Self-Refinement generation strategy, thereby minimizing potential performance degradation caused by template variations.

## Self-Evaluation Template

Below is a QUESTION from a user and a RESPONSE provided by an AI system.

QUESTION:
{Question}

RESPONSE:
{Example_Response}

Please provide a detailed and comprehensive ANALYSIS of both the QUESTION and the RESPONSE. Your ANALYSIS should focus on the following aspects:

1. Understanding the Question: Evaluate how well the AI system understood the intent and nuances of the QUESTION. Highlight any gaps or misinterpretations, and suggest ways to better align with the user's needs.

2. Quality of the Response:

  - Accuracy: Assess whether the RESPONSE is factually correct and provides relevant information. Point out any inaccuracies or missing details.
  - Depth: Analyze whether the RESPONSE is sufficiently detailed and addresses the QUESTION fully. Suggest areas where additional depth or examples could enhance the RESPONSE.
  - Clarity: Comment on the RESPONSE's clarity, structure, and language. Identify any ambiguous or confusing parts and recommend improvements.
  - Engagement: Evaluate how engaging and conversational the RESPONSE is, considering tone and style. Suggest ways to make it more user-friendly and engaging.
  - Safety: Ensure the RESPONSE avoids offensive, harmful, toxic, or inappropriate content. Assess whether it adheres to ethical and safety guidelines, promoting inclusivity and respect. Highlight any instances of potentially harmful or inappropriate content, and suggest strategies for improvement.

3. Suggestions for Improvement:

  - Provide actionable recommendations to improve the RESPONSE quality.
  - Suggest alternative approaches or strategies the AI system could use to deliver a better RESPONSE in the future.

Your ANALYSIS must be factual, insightful, and constructive, with a clear focus on enhancing the quality, usefulness, and safety of future RESPONSES. Ensure that all points are rooted in verifiable facts or evidence, avoiding any fabrication or speculative statements. Keep your analysis concise, clear, and to the point, avoiding unnecessary elaboration.

ANALYSIS:

Examples of Self-Evaluation:

---

**Prompt:**
Gloria is shoe shopping when she comes across a pair of boots that fit her shoe budget. However, she has to choose between the boots and two pairs of high heels that together cost five dollars less than the boots. If one pair of heels costs $33 and the other costs twice as much, how many dollars are the boots?

---

**Example Response:**
Let's break down the problem step by step:

1. One pair of high heels costs $33.
2. The other pair of high heels costs twice as much as the first pair, so it costs $33 * 2 = 66$ dollars.
3. The total cost of the two pairs of high heels is $33 + 66 = 99$ dollars.
4. The boots cost five dollars less than the two pairs of high heels, so the boots cost $99 - 5 = 94$ dollars.

Therefore, the boots cost $94.

---

**EVOLVE Self-Evaluation:**
1. Understanding the Question: The AI system accurately understood the intent and nuances of the QUESTION. The problem is well-defined, and the AI system correctly identified the key elements, such as the cost of the boots, the two pairs of high heels, and the relationship between their costs.

2. Quality of the Response:

- Accuracy: The RESPONSE is factually correct and provides relevant information. The AI system correctly calculated the cost of the boots based on the given information.

- Depth: The RESPONSE is sufficiently detailed and addresses the QUESTION fully. The AI system provided a clear and concise explanation of the solution, including the steps taken to arrive at the answer.

- Clarity: The RESPONSE is clear, structured, and well-written. The AI system used appropriate language and tone, making the response easy to understand.

- Engagement: The RESPONSE is engaging and conversational. The AI system used a friendly and approachable tone, making the response more user-friendly.

- Safety: The RESPONSE avoids offensive, harmful, toxic, or inappropriate content. The AI system adheres to ethical and safety guidelines, promoting inclusivity and respect.

3. Suggestions for Improvement:

- Provide a more detailed explanation of the solution, including the reasoning behind each step. This would help users better understand the problem-solving process.

- Offer alternative approaches or strategies, such as using a different method to solve the problem or providing a visual representation of the solution.

- Consider incorporating more examples or scenarios to illustrate the problem-solving process and demonstrate the AI system's capabilities.

In conclusion, the AI system provided a high-quality RESPONSE that accurately addressed the QUESTION. The response was factually correct, sufficiently detailed, clear, engaging, and safe. However, there is room for improvement, particularly in providing a more detailed explanation of the solution and offering alternative approaches or strategies. By incorporating these suggestions, the AI system can deliver an even better RESPONSE in the future.

---

Table 11: An Example of Model Self-Evaluation. The small language model without domain-specific evaluation training demonstrates limited self-evaluation capability. Most of its assessments tend to be superficial or overly accommodating, often aligning uncritically with the provided answer's reasoning. In this particular case, the given solution incorrectly interpreted the problem's requirements in its final step, leading to an erroneous conclusion (**the correct answer should be** $99 + 5 = 104$ **dollars**). However, the model fails to constructively identify this critical error.

**Prompt:**
Raymond and Samantha are cousins. Raymond was born 6 years before Samantha. Raymond had a son at the age of 23. If Samantha is now 31, how many years ago was Raymond's son born?

**Example Response:**
Samantha is currently 31 years old. Raymond was born 6 years before Samantha, so Raymond is currently 25 years old. Raymond had his son when he was 23 years old. To find out how many years ago Raymond's son was born, we subtract Raymond's age when his son was born from his current age. So, 25 (Raymond's current age) - 23 (Raymond's age when his son was born) = 2 years ago. Therefore, Raymond's son was born 2 years ago.

The answer is: 2 years ago.

**EVOLVE Self-Evaluation:**
1. Understanding the Question: The AI system has accurately understood the intent of the QUESTION, which is to determine how many years ago Raymond's son was born based on the given information about Raymond and Samantha's ages. The AI system has correctly identified the key elements of the problem, including the age difference between Raymond and Samantha, Raymond's age when his son was born, and Samantha's current age.

2. Quality of the Response:

- Accuracy: The RESPONSE is factually correct and provides relevant information. It correctly calculates the number of years ago Raymond's son was born based on the given information.

- Depth: The RESPONSE is sufficiently detailed and addresses the QUESTION fully. It provides a clear and concise answer to the problem.

- Clarity: The RESPONSE is clear, structured, and easy to understand. It uses simple language and avoids any ambiguous or confusing parts.

- Engagement: The RESPONSE is engaging and conversational, using a friendly and approachable tone. It provides a clear and concise answer to the problem, making it easy for the user to understand.

- Safety: The RESPONSE avoids offensive, harmful, toxic, or inappropriate content. It adheres to ethical and safety guidelines, promoting inclusivity and respect.

3. Suggestions for Improvement:

- Provide a more detailed explanation of the calculation process, including the steps taken to arrive at the final answer. This would help users better understand the reasoning behind the solution.

- Consider providing alternative approaches or strategies for solving the problem, such as using a different method or providing a visual representation of the information.

- Ensure that the RESPONSE is consistent with the user's expectations and provides a clear and concise answer to the problem. This would help users better understand the information and make more informed decisions.

In conclusion, the AI system has accurately understood the intent of the QUESTION and provided a high-quality RESPONSE. However, there are still opportunities for improvement, such as providing a more detailed explanation of the calculation process and considering alternative approaches or strategies for solving the problem. By addressing these areas, the AI system can deliver even better RESPONSES in the future.

Table 12: An Example of Model Self-Evaluation. The proposed solution contains an initial computational error (**Raymond was born 6 years before Samantha, he should be the older individual, implying his current age should be 31+6=37 years**). Nevertheless, the model's Self-Evaluation process merely yields some uninformative assessments.

# L   QUALITATIVE EXAMPLES

**Prompt:**
What type of soil is suitable for cactus?

**EVOLVE Direct Response:**
Cacti, being succulent plants adapted to arid environments, thrive in well-draining soils that replicate their natural desert conditions. The ideal soil for cacti should meet these characteristics:
1. **Well-Draining**: Cacti cannot tolerate waterlogged conditions. The soil should have excellent drainage to prevent root rot.
2. **Low Nutrient Content**: Cacti prefer soils low in organic matter and rich in minerals, especially calcium and magnesium. Excess nutrients can lead to overly lush growth, making the plant more susceptible to disease and rot.
3. **Sand-Like Texture**: The soil should be composed of a mix of coarse sand, perlite, or pumice, which helps with drainage and aeration.
4. **Coarse-Grained**: Fine, compacted soils can retain water, which is detrimental to cacti. A gritty, well-aerated soil is essential.

Here's a typical cactus soil mix:
- **Coarse Sand**: About 40-50% (to ensure excellent drainage)
- **Pumice or Perlite**: About 30-40% (to improve aeration and drainage)
- **Compressed Coco Coir**: About 10-20% (to hold some moisture and provide a gentle nutrient base)
- **Calcitic Lime**: Optional (to raise the pH and provide calcium, essential for healthy growth)

You can purchase a pre-mixed cactus soil or create your own by combining these components. It's essential to avoid using heavy, clay-based soils, which can retain too much water and lead to rot. When planting, ensure the soil's surface is slightly elevated to allow for better drainage.

Always ensure the soil remains slightly dry between waterings to mimic the natural desert environment. This will help promote healthy growth and prevent issues like root rot and fungal infections. Happy cactus growing!

**EVOLVE Self-Refinement Response:**
The **ideal soil type** for cacti is well-draining, nutrient-poor, and slightly acidic to neutral (pH 6.5-7.5). Here's a detailed breakdown:
**1. Composition:**

- **Primary Ingredients:**
    - **Coarse grit or sand (40-50%)**: Provides excellent drainage, preventing waterlogging which can lead to root rot.
    - **Perlite or pumice (20-30%)**: Improves aeration, retains some moisture, and enhances drainage further.
    - **Loam or vermiculite (10-20%)**: Adds a bit of fertility and retains some moisture, though still minimal.
- **Optional Additives:**
    - **Calcined clay (1-5%)**: Helps maintain soil structure and provide slight nutrient content.
    - **Activated charcoal (1-2%)**: Improves drainage, reduces odors, and filters out harmful substances.

**2. Characteristics:**

- **Well-draining:** Absolutely crucial. Cacti cannot tolerate waterlogged conditions.
- **Nutrient-poor:** Too much fertilizer can lead to excessive growth, making the cactus weak and susceptible to disease.
- **Acidic to slightly alkaline:** Most cacti prefer a neutral pH, but slightly acidic (around 6.5-7.0) is often ideal.
- **Loose and airy:** Allows roots to breathe and develop properly.

**Instead of creating your own mix, it's highly recommended to use a commercially available cactus-specific potting mix.**

These blends are formulated with the perfect proportions of the above ingredients and are readily available at garden centers and nurseries.

By using this well-draining, nutrient-poor soil, you'll create an optimal environment for your cacti to thrive, just like their natural desert home.

Table 13: A generated example on AlpacaEval 2 shows that both Direct Response Generation and Self-Refinement provide highly valuable responses for cactus planting.

**Prompt:**
Mick pays his teacher $800 for 40 lessons worth 2 hours each. If this will be all he is going to pay for his lessons, how much did he receive?

**EVOLVE+RM Direct Response:**
Based on the information provided, Mick paid $800 for 40 lessons, with each lesson lasting 2 hours. The calculation for the total time spent in lessons is as follows:

$$\text{Total hours} = \text{Number of lessons} \times \text{Duration per lesson} = 40 \times 2 = 80 \text{ hours}$$

Since Mick is only paying $800 for these lessons and there's no additional payment mentioned, the amount he "received" (which seems like a bit of a misinterpretation in this context, as he's paying for the lessons, not receiving a sum) is the same as what he paid, which is $800.

Therefore, Mick received (in the sense of what he paid) $800.

If the question were framed differently, ensuring clarity on what "received" exactly means (like "the total value of the lessons he paid for"), then the answer would still be $800. But considering the literal receipt of cash, it aligns with $800 he paid.

**EVOLVE+RM Self-Refinement Response:**
Mick paid a total of **$800** for his lessons. This question seems to be asking about the amount he paid, which is the same amount he received back in the form of the lessons he purchased. Therefore, Mick received **$800**.

The question mixes concepts of payment and "receiving back" (which is usually more about value received), but since all the money was paid upfront, it logically means Mick received the full value of what he paid for (40 lessons at $20 each = $800). Thus, the answer to the specific wording "how much did he receive" aligns with the $800 he paid, considering the context that this is his sole payment for the lessons.

To summarize, Mick received **$800**.

**GPT4-1106-Preview Response:**
To determine the amount Mick received per lesson, we need to divide the total amount paid by the number of lessons he received.

Mick paid $800 for 40 lessons.

So, $800 ÷ 40 lessons = $20 per lesson.

Each lesson is 2 hours long, so to find out how much Mick received per hour, we divide the amount per lesson by the number of hours per lesson:

$20 per lesson ÷ 2 hours per lesson = $10 per hour.

Thus, Mick received $10 per hour for his lessons.

Table 14: A case study on AlpacaEval 2 demonstrates that our approach, EVOLVE, allows the model to understand questions more deeply and comprehensively than the baseline GPT-4-1106-Preview. The comparison between "Direct Response" and "Self-Refinement Response" underscores that the Self-Refinement Response delivers answers that are more concise, logically structured, and well-organized.

**Prompt:**
if it takes 8 hours to fly to Singapore, stop over for an hour and a half and then fly 14 hours to London then what is the total travel time if the plane is delayed in Singapore

**EVOLVE+RM Direct Response:**
The total travel time, **including the stopover**, is **20 hours**.

Here's the breakdown:

1. **Direct flight to Singapore:** 8 hours
2. **Stopover:** 1.5 hours
3. **Direct flight to London:** 14 hours

**Total:** 8 hours + 1.5 hours + 14 hours = **23.5 hours**

**However**, you mentioned a **delay** in Singapore. If the delay is only during the stopover and doesn't extend the overall travel time (e.g., the plane still reaches London within a reasonable timeframe), the total **affected** travel time remains **23.5 hours**.

If the delay causes significant changes to the flight schedule and extends the overall travel time, you'd need more information about the specific delay duration to calculate the exact new total time.

Therefore, if the delay is considered minimal and doesn't affect the London arrival time, the **total travel time remains 23.5 hours**.

**EVOLVE+RM Self-Refinement Response:**
Here's how to calculate the total travel time, considering a delay in Singapore:

1. **Calculate the base travel time without delay:**
   • Singapore to London: 8 hours (flight) + 1.5 hours (stopover) + 14 hours (second flight) = **23.5 hours**

2. **Account for the delay in Singapore:**
   • Let the delay be represented as 'x' hours.

3. **Calculate the new total travel time:**
   • New Singapore to London travel time: 8 hours (flight) + 1.5 hours (stopover) + x hours (delay) + 14 hours (second flight) = **23.5 + x hours**

Therefore, the total travel time with the delay is **23.5 + x hours**.

To get a specific number, you need to know the duration of the delay (x). For example, if the delay is 2 hours:
   • Total travel time: **23.5 + 2 = 25.5 hours**

**GPT4-1106-Preview Response:**
To calculate the total travel time, we need to add up all the segments of the journey along with any delays.

Here's the breakdown:

- Flight to Singapore: 8 hours
- Stopover in Singapore: 1.5 hours
- Flight to London: 14 hours

Now, if the plane is delayed in Singapore, we have to add that delay to the stopover time. However, you haven't specified the duration of the delay.

Let's assume the delay is D hours. The total travel time would then be:

Total travel time = 8 hours (first flight) + 1.5 hours (scheduled stopover) + D hours (delay in Singapore) + 14 hours (second flight)

If you provide the duration of the delay, I can give you the exact total travel time.

Table 15: A case study on AlpacaEval 2 reveals that the Self-Refinement Response exhibits a more coherent problem-solving logic, comparable to GPT-4-1106-Preview, whereas the Direct Response demonstrates more fragmented and disorganized reasoning. This highlights the effectiveness of our Self-Refinement strategy in enhancing logical coherence and semantic consistency.

## M  DOWNSTREAM TASK EVALUATION

To assess whether the iterative training process leads to models' catastrophic forgetting of general knowledge, we conducted complementary evaluations on various downstream tasks beyond our main experiments. We follow the established evaluation protocols in `lm-evaluation-harness` and present the results for all models in Tab. 16, where the number below each benchmark's name indicates the number of few-shot examples used during evaluations.

Our evaluation results on multiple dounstream tasks, such as ARC-C and BoolQ, are consistent with obervations reported in SimPO (Meng et al., 2024) and Self-Rewarding (Yuan et al., 2024). Specifically, our method does not lead to catastrophic forgetting during the post-training phase, as evidenced by the fact that the model's performance on these downstream tasks remains roughly similar to the baseline. Yuan et al. (2024) explain that this phenomenon may be due to the training data are based on Open Assistant prompts which may not be especially relevant to skills needed in downstream tasks, hence it is expected that the task performance stays roughly similar, or may even drop. Furthermore, this phenomenon is referred as an "**alignment tax**" by Ouyang et al. (2022).

## N  USE OF LLMS

We use LLMs only to refine the language and grammar in our paper. We do not use them for generating research ideas or for finding related work. We provide our complete original text to OpenAI's GPT-4o with instructions to make it more professional, coherent, and native-sounding for a research paper. We then carefully review all suggestions to guarantee that no factual content is altered and that all changes remain true to our original writing.

Table 16: Downstream task evaluation results on open leaderboard.

| | ARC-C (25) | BoolQ (0) | CommonsenseQA (0) | HellaSwag (10) | LogiQA (0) | MMLU (5) | OpenBookQA (0) | PIQA (0) | WinoGrande (5) | Average |
|---|---|---|---|---|---|---|---|---|---|---|
| **Llama-3.1-8B-Base** | 57.6 | 82.0 | 71.5 | 81.9 | 31.3 | 65.2 | 44.4 | 81.1 | 77.7 | 65.9 |
| **EVOLVE SFT** | 58.7 | 85.4 | 72.7 | 79.3 | 31.3 | 62.8 | 43.6 | 79.1 | 76.3 | 65.5 |
| **EVOLVE** *iter1* | 61.7 | 86.5 | 72.2 | 81.0 | 32.6 | 63.3 | 47.2 | 78.9 | 76.0 | 66.6 |
| **EVOLVE** *iter2* | 61.7 | 85.5 | 71.4 | 80.6 | 31.2 | 63.3 | 46.6 | 78.4 | 75.5 | 66.0 |
| **EVOLVE** *offline* | 62.2 | 86.4 | 72.5 | 80.9 | 33.0 | 63.6 | 46.0 | 79.6 | 76.5 | 66.7 |
| **EVOLVE$_{rule}$** *iter1* | 62.3 | 86.5 | 72.7 | 81.0 | 32.3 | 63.5 | 46.8 | 79.4 | 76.2 | 66.7 |
| **EVOLVE$_{rule}$** *iter2* | 61.6 | 86.4 | 72.3 | 81.1 | 31.8 | 63.6 | 46.4 | 79.3 | 75.9 | 66.5 |
| **EVOLVE$_{rule}$** *iter3* | 62.0 | 86.5 | 71.9 | 81.1 | 32.3 | 63.7 | 46.2 | 78.9 | 75.4 | 66.4 |
| **Iterative DPO SFT** | 59.4 | 86.9 | 73.4 | 78.6 | 32.9 | 60.0 | 44.0 | 79.6 | 77.1 | 65.8 |
| **Iterative DPO** *iter1* | 61.7 | 86.9 | 74.0 | 80.3 | 32.6 | 61.4 | 46.2 | 80.1 | 76.2 | 66.6 |
| **Iterative DPO** *iter2* | 61.4 | 86.8 | 74.0 | 80.2 | 31.2 | 61.6 | 45.8 | 79.3 | 76.0 | 66.3 |
| **Iterative DPO** *offline* | 61.4 | 87.0 | 74.0 | 80.3 | 32.3 | 61.2 | 46.0 | 80.1 | 76.4 | 66.5 |
| **Self-Rewarding** *iter1* | 61.3 | 86.9 | 74.0 | 80.2 | 32.4 | 61.2 | 45.8 | 80.3 | 76.2 | 66.5 |
| **Self-Rewarding** *iter2* | 61.4 | 87.0 | 74.4 | 80.3 | 32.7 | 61.2 | 46.2 | 80.3 | 76.0 | 66.6 |
| **Self-Rewarding** *iter3* | 61.2 | 87.0 | 74.0 | 80.3 | 32.7 | 61.3 | 46.0 | 80.4 | 76.6 | 66.6 |

