# OpenReview forum: "Evolving LLMs' Self-Refinement Capability via Synergistic Training-Inference Optimization"
_ICLR.cc/2026/Conference — ICLR 2026 Conference Withdrawn Submission_

### Official Review · Reviewer_fGNn · 2025-10-20

**Soundness:** 2
**Presentation:** 3
**Contribution:** 2
**Rating:** 2
**Confidence:** 4

**Summary:**

The paper propsoe EVOLVE, an iterative-training-based algorithm that focus on training LLM to enable the self-reflection capability. The authors evaluate the performance of the proposed algorithm on Llama and Mistral models, and show that EVOLVE can make the model stronger compared to iterative DPO.

**Strengths:**

- The paper is overall well-written and easy to follow.
- The proposed algorithm works for non-Qwen models, which is relatively unique in recent days.
- Detailed ablation study is provided to support the proposed algorithm.

**Weaknesses:**

- The base model and tested dataset are relatively weak. No difficult dataset like AIME, and Olympiad-bench are tested.
- Baselines are relatively weak. While a list of different algorithms are used, e.g., Iterative DPO, SRPO. No popular algorithms like GRPO / DAPO / GSPO are compared. This is very important since Deepseek-R1 has already shown that self-reflection capability can be learned directly through these algorithms.
- While the claim is on enabling self-reflection capability of the LLMs, no further discussion are provided to verify how much self-reflection capability are improved. Only the final accuracy on test benchmark are tested.
- Because the proposed algorithms are based on manually creating the preference-based optimization, the general applicability to different domains needs further examinations. While Arena-hard is a comprehensive benchmark, details breakdown are needed.

**Questions:**

While I recommend the authors to refer to the weakness points I listed, the two primary question I have is:
- Can you provide some comparison to GRPO-like RL algorithms?
- Can you provide some discussion on how much self-reflection capability are improved compared to baselines?

---

### Official Review · Reviewer_jY17 · 2025-10-31

**Soundness:** 3
**Presentation:** 3
**Contribution:** 3
**Rating:** 8
**Confidence:** 3

**Summary:**

The authors address the problem of the shortage of high-quality training data by focusing on self-improvement, in which a model enhances its own capability using data it generates. However, the effectiveness of self-improvement depends on the quality of the synthetic data produced by the model itself, which in turn requires improving output accuracy and stability. To this end, the authors propose leveraging SR (Self-Refinement), a process in which the model revises its own responses during inference, to enhance these properties.
Since current LLMs do not naturally possess strong self-refinement ability, as confirmed experimentally, the authors propose a two-stage framework called EVOLVE: in the first stage, the model acquires SR capability through SFT; in the second, it performs self-improvement using the high-quality synthetic data generated through SR.

**Strengths:**

- The performance is good.
- They introduce a new loss function to optimize the policy for enabling self-correction.
- They explore four different ways to use SR capability during inference. The comparison among the four patterns is interesting. ((it also feels somewhat expected, as Chain of Self-Refinement follows a process similar to the training distribution. I believe there is still room for exploration in how SR is applied at inference time. For example, from a test-time scaling perspective, parallel sampling with majority voting is known to improve accuracy as the number of samples increases, so combining this with Chain of Self-Refinement might further enhance performance).

**Weaknesses:**

- Only small models (7B, 8B) are tested. It would be even better if experiments were conducted on larger models.

- In Figure 1, the authors use Qwen2.5-7B and Gemma 2-9B, but these models are not used in Section 4.1. Why? It would be preferable to conduct a more comprehensive set of experiments covering all models.

**Questions:**

- Why did the authors stop the iterations at three? The trend suggests that performance could keep increasing; does accuracy fail to improve beyond this point?

---

### Official Review · Reviewer_pzNe · 2025-11-01

**Soundness:** 3
**Presentation:** 3
**Contribution:** 3
**Rating:** 4
**Confidence:** 3

**Summary:**

This paper studies the intrinsic self-refinement ability of LLMs: the ability to iteratively revise and improve their own outputs without additional supervision. The authors show via preliminary experiments that current LLMs do not reliably self-refine, and instead often degrade quality over iterations. To address this, they propose EVOLVE, a two-stage framework combining (1) Self-Refinement-aware SFT to activate refinement capability and (2) iterative preference training (PT) to strengthen refinement across rounds. At inference time, chain-of-self-refinement generation is used to iteratively improve responses and generate synthetic preference data, which is used to further train the model in a closed loop. EVOLVE yields substantial gains, enabling a Llama-3.1-8B base model to surpass GPT-4o on AlpacaEval2 and achieve strong Arena-Hard win rates. It also generalizes to out-of-domain math tasks, improving performance on GSM8K and MATH benchmarks

**Strengths:**

- The paper addresses a central open question in self-improving LLMs: Do models inherently self-refine? The empirical finding, they do not, is significant and timely.
- It is a comprehensive, end-to-end solution that correctly identifies that a new capability requires both training and a mechanism to apply it. It also identifies and formalizes self-refinement activation and training.
- The empirical results seem to be quite strong.

**Weaknesses:**

- While results on AlpacaEval2/Arena-Hard are strong, evaluation is largely in general-instruction domains. Math results are promising but limited. The paper could use an inclusion of a broader set (coding, safety, knowledge-intensive QA) to test if self-refinement holds across modalities and task difficulty.
- EVOLVE uses reward-model scoring to build preference data. Reward-model bias or reward hacking is lightly discussed but not fully quantified. Maybe the authors can report RM perplexity drift or diversity metrics across iterations.
- Iterative multi-round response generation (4 rounds x 5–10k prompts/iter) may be costly. The authors could also report compute cost vs performance curves.
- The paper acknowledges instability in alternative self-improvement baselines, but failure cases for EVOLVE are not deeply analyzed.

**Questions:**

See weaknesses.

---

### Official Review · Reviewer_oujH · 2025-11-12

**Soundness:** 2
**Presentation:** 1
**Contribution:** 3
**Rating:** 2
**Confidence:** 4

**Summary:**

The paper studies LLMs' ability of "self refinement ", ie, a model's ability to revise its own responses without external information and the effect on reasoning evaluations.
self refinement is studied both as an inference time augmentation, as well as a strategy for iterative training to improve (zero shot) responses (referred to as "self improvement"), and to improve the ability of the trained model to self refine answers.

The central question of the paper is whether LLMs can autonomously refined their responses without external information.
The paper starts out with an evaluation that answers this in the negative for existing open weights 7-9B LLMs. Then it proceeds to study and suggest a number of different training strategies, which improve self refinement (from a negative impact to a neutral-to-slightly positive), and additionally substantially increase zero shot performance, and downstream performance on GSM8K and MATH.

**Strengths:**

- the paper studies and interesting problem of improving "self refinement", which can be thought of as an important form of reasoning for LLMs.
- the evaluation set up follows a well established method, which is similar to e.g. Self-Rewarding Language Models (Yuan et al. ICML 2024, https://openreview.net/pdf?id=0NphYCmgua) and Iterative Reasoning Preference Optimization (Pang et al, Neurips 2024 -- paper cites the archive pre-print by Yuan et al. only).
- the paper surfaces a significant reasoning limitation of 7-9B LLMs, ie, that these models are not able to refine their own answers productively.
- the paper studies many different variants of training and inference approaches.
- It shows that using an external preference model and iterative training on model outputs improves model performance (zero shot) and generalizes to GSM8K and MATH datasets.

**Weaknesses:**

## Self-Refinement vs Self-Improvement
- I think the paper should define more clearly, what is meant by self refinement versus self improvement. I think that the authors mean that self refinement is an inference time procedure, while self improvement means using the model's generations in additional training.
- While existing models show a roughly neutral to strongly negative impact when using self refinement, the methods proposed in this paper do not result in models which can significantly refine their answers with increased self refinement inference turns either (cf.  flat curves over inference turns in fig. 3, 6a, 9a). However the zero shot abilities for model responses improve with the number of training iterations.
I believe that this result somewhat undermines the papers focus on "evolving LLMs self refinement capability" (line 0).


## Claims
- Line 84: the summary of the contributions states that the paper empirically validate the effectiveness of the framework in enhancing self refinement capability.
- Line 193 claims that figure 3 demonstrates that the two training phases (SFT + PT) are truly complementary. I don't find this claim substantiated in figure 3, which shows different combination of SFT + PT choices, but not either alone.
- Line 44 States that the main research question is whether LLMs can autonomously refine their responses without external information. While there is nuance,

## Paper presentation:
The paper is poorly written and the presentation makes it difficult to understand, in particular the following points.
### 1. Language clarity:
the first thing that jumps out on the first few pages, and continues throughout, is a very liberal use of adjectives and formulations with meanings which are not defined in the text nor clear to me. It seems apparent that the stated use of LLM's in writing. This paper has worked against the intended effect of increased clarity:
- line 2: "synergistic" training-inference optimization
- line 17: "inherent" self-refinement
- line 21: "broader" self-improvement of intrinsic model abilities.
- line 22: "evolved" self-refinement ability
- line 69: talks about the "activation and strengthening " of self refinement
- line 69: "sustained" self improvement
- line 73 "activated" self refinement capability
- line 77: "eliciting and enhancing" self refinement

### 2. Differentiation in related works:
This paper is most similar to the above mentioned works by Yuan et al. and Pang et al., which extends even to the color scheme and layout of figure 2, and line 100+ establishes differentiation from them with terms that are not clearly defined in the paper:
> "our work builds on these foundations, focusing on activating and enhancing self refinement as a mechanism for sustained self improvement, distinct from prior approaches by emphasizing iterative training to strengthen intrinsic refinement capabilities."

### 3. Paper structure
I think the structure of the paper should be improved by moving content into their respective sections:
- the introduction contains the results presented in figure 1.
- the method section contains the results and figure 3.

**Questions:**

My main concerns and questions were stated above.

**Details Of Ethics Concerns:**

-

---

### Note · Authors · 2026-01-04

I have read and agree with the venue's withdrawal policy on behalf of myself and my co-authors.